# INFLUENCE-BASED MULTI-AGENT EXPLORATION

**Tonghan Wang**[*]**, Jianhao Wang**[*]**, Yi Wu & Chongjie Zhang**
Institute for Interdisciplinary Information Sciences
Tsinghua University
Beijing, China
`wangth18@mails.tsinghua.edu.cn`, `wjh720.eric@gmail.com`
`jxwuyi@openai.com`, `chongjie@tsinghua.edu.cn`

## ABSTRACT

Intrinsically motivated reinforcement learning aims to address the exploration challenge for sparse-reward tasks. However, the study of exploration methods in transition-dependent multi-agent settings is largely absent from the literature. We aim to take a step towards solving this problem. We present two exploration methods: exploration via information-theoretic influence (EITI) and exploration via decision-theoretic influence (EDTI), by exploiting the role of interaction in coordinated behaviors of agents. EITI uses mutual information to capture the interdependence between the transition dynamics of agents. EDTI uses a novel intrinsic reward, called Value of Interaction (VoI), to characterize and quantify the influence of one agent's behavior on expected returns of other agents. By optimizing EITI or EDTI objective as a regularizer, agents are encouraged to coordinate their exploration and learn policies to optimize the team performance. We show how to optimize these regularizers so that they can be easily integrated with policy gradient reinforcement learning. The resulting update rule draws a connection between coordinated exploration and intrinsic reward distribution. Finally, we empirically demonstrate the significant strength of our methods in a variety of multi-agent scenarios.

## 1 INTRODUCTION

Reinforcement learning algorithms aim to learn a policy that maximizes the accumulative reward from an environment. Many advances of deep reinforcement learning rely on a dense shaped reward function, such as distance to the goal (Mirowski et al., 2016; Wu et al., 2018), scores in games (Mnih et al., 2015) or expert-designed rewards (Wu & Tian, 2016; OpenAI, 2018), but they tend to struggle in many real-world scenarios with sparse rewards (Burda et al., 2019). Therefore, many recent works propose to introduce additional intrinsic incentives to boost exploration, including pseudo-counts (Bellemare et al., 2016; Tang et al., 2017; Ostrovski et al., 2017), model-learning improvements (Burda et al., 2019; Pathak et al., 2017; Burda et al., 2018), and information gain (Florensa et al., 2017; Gupta et al., 2018; Hyoungseok Kim, 2019). These works result in significant progress in many challenging tasks such as Montezuma Revenge (Burda et al., 2018), robotic manipulation (Pathak et al., 2018; Riedmiller et al., 2018), and Super Mario games (Burda et al., 2019; Pathak et al., 2017).

Notably, most of the existing breakthroughs on sparse-reward environments have been focusing on single-agent scenarios and leave the exploration problem largely unstudied for multi-agent settings – it is common in real-world applications that multiple agents are required to solve a task in a coordinated fashion (Cao et al., 2012; Nowé et al., 2012; Zhang & Lesser, 2011). This problem has recently attracted attention and several exploration strategies have been proposed for transition-independent cooperative multi-agent settings (Dimakopoulou & Van Roy, 2018; Dimakopoulou et al., 2018; Bargiacchi et al., 2018; Iqbal & Sha, 2019b). Nevertheless, how to explore effectively in more general scenarios with complex reward and transition dependency among cooperative agents remains an open research problem.

---

[*]Equal Contribution.

This paper aims to take a step towards this goal. Our basic idea is to coordinate agents' exploration by taking into account their interactions during their learning processes. Configurations where interaction happens (interaction points) lie at critical junctions in the state-action space, through these critical configurations can transit to potentially important under-explored regions. To exploit this idea, we propose exploration strategies where agents start with decentralized exploration driven by their individual curiosity, and are also encouraged to visit interaction points to influence the exploration processes of other agents and help them get more extrinsic and intrinsic rewards. Based on how to quantify influence among agents, we propose two exploration methods. *Exploration via information-theoretic influence* (EITI) uses mutual information (MI) to capture the interdependence between the transition dynamics of agents. *Exploration via decision-theoretic influence* (EDTI) goes further and uses a novel measure called *value of interaction* (VoI) to disentangle the effect of one agent's state-action pair on the expected (intrinsic) value of other agents. By optimizing MI or VoI as a regularizer to the value function, agents are encouraged to explore state-action pairs where they can exert influences on other agents for learning sophisticated multi-agent cooperation strategies.

To efficiently optimize MI and VoI, we propose augmented policy gradient formulations so that the gradients can be estimated purely from trajectories. The resulting update rule draws a connection between coordinated exploration and the distribution of individual intrinsic rewards among team members, which further explains why our methods are able to facilitate multi-agent exploration.

We demonstrate the effectiveness of our methods on a variety of sparse-reward cooperative multi-agent tasks. Empirical results show that both EITI and EDTI allow for the discovery of influential states and EDTI further filter out interactions that have no effects on the performance. Our results also imply that these influential states are implicitly discovered as subgoals in search space that guide and coordinate exploration. The video of experiments is available at `https://sites.google.com/view/influence-based-mae/`.

## 2 SETTINGS

In our work, we consider a fully cooperative multi-agent task that can be modelled by a factored multi-agent MDP $G = \langle N, S, A, T, r, h, n \rangle$, where $N \equiv \{1, 2, ..., n\}$ is the finite set of agents, $S \equiv \times_{i \in N} S_i$ is the finite set of joint states and $S_i$ is the state set of agent $i$. At each timestep, each agent selects an action $a_i \in A_i$ at state $s$, forming a joint action $\boldsymbol{a} \in A \equiv \times_{i \in N} A_i$, resulting in a shared extrinsic reward $r(\boldsymbol{s}, \boldsymbol{a})$ for each agent and the next state $\boldsymbol{s}'$ according to the transition function $T(\boldsymbol{s}'|\boldsymbol{s}, \boldsymbol{a})$.

The objective of the task is that each agent learns a policy $\pi_i(a_i|s_i)$, jointly maximizing team performance. The joint policy $\boldsymbol{\pi} = \langle \pi_1, \ldots, \pi_n \rangle$ induces an action-value function, $Q^{ext,\boldsymbol{\pi}}(\boldsymbol{s}, \boldsymbol{a}) = \mathbb{E}_\tau[\sum_{t=0}^h r^t | \boldsymbol{s}^0 = \boldsymbol{s}, \boldsymbol{a}^0 = \boldsymbol{a}, \boldsymbol{\pi}]$, and a value function $V^{ext,\boldsymbol{\pi}}(\boldsymbol{s}) = \max_{\boldsymbol{a}} Q^{ext,\boldsymbol{\pi}}(\boldsymbol{s}, \boldsymbol{a})$, where $\tau$ is the episode trajectory and $h$ is the horizon.

We adopt a centralized training and decentralized execution paradigm, which has been widely used in multi-agent deep reinforcement learning (Foerster et al., 2016; Lowe et al., 2017; Foerster et al., 2018; Rashid et al., 2018). During training, agents are granted access to the states, actions, (intrinsic) rewards, and value functions of other agents, while decentralized execution only requires individual states.

## 3 INFLUENCE-BASED COORDINATED MULTI-AGENT EXPLORATION

Efficient exploration is critical for reinforcement learning, particularly in sparse-reward tasks. Intrinsic motivation (Oudeyer & Kaplan, 2009) is a crucial mechanism for behaviour learning since it provides the driver of exploration. Therefore, to trade off exploration and exploitation, it is common for an RL agent to maximize an objective of the expected extrinsic reward augmented by the expected intrinsic reward. Curiosity is one of the extensively-studied intrinsic rewards to encourage an agent to explore according to its uncertainty about the environment, which can be measured by model prediction error (Burda et al., 2019; Pathak et al., 2017; Burda et al., 2018) or state visitation count (Bellemare et al., 2016; Tang et al., 2017; Ostrovski et al., 2017).

While such an intrinsic motivation as curiosity drives effective individual exploration, it is often not sufficient enough for learning in collaborative multi-agent settings, because it does not take

into account agent interactions. To encourage interactions, we propose an influence value aims to quantify one agent's influence on the exploration processes of other agents. Maximizing this value will encourage agents to visit interaction points more often through which the agent team can reach configurations that are rarely visited by decentralized exploration. In next sections, we will provide two ways to formulate the influence value with such properties, leading to two exploration strategies.

Thus, for each agent $i$, our overall optimization objective is:

$$J_{\theta_i}[\pi_i|\pi_{-i}, p_0] \equiv V^{ext,\boldsymbol{\pi}}(\boldsymbol{s}_0) + V_i^{int,\boldsymbol{\pi}}(\boldsymbol{s}_0) + \beta \cdot I_{-i|i}^{\boldsymbol{\pi}}, \tag{1}$$

where $p_0(\boldsymbol{s}_0)$ is the initial state distribution, $\pi_{-i}$ is the joint policy excluding that of agent $i$, and $V_i^{int,\boldsymbol{\pi}}(\boldsymbol{s})$ is the intrinsic value function of agent $i$, $I_{-i|i}^{\boldsymbol{\pi}}$ is the influence value, $\beta > 0$ is a weighting term. In this paper, we use the following notations:

$$\tilde{r}_i(\boldsymbol{s}, \boldsymbol{a}) = r(\boldsymbol{s}, \boldsymbol{a}) + u_i(s_i, a_i), \tag{2}$$

$$V_i^{\boldsymbol{\pi}}(\boldsymbol{s}) = V^{ext,\boldsymbol{\pi}}(\boldsymbol{s}) + V_i^{int,\boldsymbol{\pi}}(\boldsymbol{s}), \tag{3}$$

$$Q_i^{\boldsymbol{\pi}}(\boldsymbol{s}, \boldsymbol{a}) = \tilde{r}_i(\boldsymbol{s}, \boldsymbol{a}) + \sum_{\boldsymbol{s}'} T(\boldsymbol{s}'|\boldsymbol{s}, \boldsymbol{a}) V_i^{\boldsymbol{\pi}}(\boldsymbol{s}'), \tag{4}$$

where $u_i(s_i, a_i)$ is a curiosity-derived intrinsic reward, $\tilde{r}_i(\boldsymbol{s}, \boldsymbol{a})$ is a sum of intrinsic and extrinsic rewards, $V_i^{\boldsymbol{\pi}}(\boldsymbol{s})$ and $Q_i^{\boldsymbol{\pi}}(\boldsymbol{s}, \boldsymbol{a})$ here contain both intrinsic and extrinsic rewards.

## 3.1 Exploration via Information-Theoretic Influence

One critical problem in our learning framework presented above is to define the influence value $I$. For simplicity, we start with a two-agent case. The first method we propose is to use mutual information between agents' trajectories to measure one agent's influence on other agents' learning processes. Such mutual information can be defined as information gain of one agent's state transition given the other's state and action. Without loss of generality, we define it from the perspective of agent 1:

$$MI_{2|1}^{\boldsymbol{\pi}}(S_2'; S_1, A_1|S_2, A_2) = \sum_{\boldsymbol{s}, \boldsymbol{a}, s_2' \in (S, A, S_2)} p^{\boldsymbol{\pi}}(\boldsymbol{s}, \boldsymbol{a}, s_2') \left[\log p^{\boldsymbol{\pi}}(s_2'|\boldsymbol{s}, \boldsymbol{a}) - \log p^{\boldsymbol{\pi}}(s_2'|s_2, a_2)\right], \tag{5}$$

where $\boldsymbol{s} = (s_1, s_2)$ is the joint state, $\boldsymbol{a} = (a_1, a_2)$ is the joint action, and $S_i$ and $A_i$ are the random variables of state and action of agent $i$ subject to the distribution induced by the joint policy $\boldsymbol{\pi}$. So we define $I_{2|1}^{\boldsymbol{\pi}}$ as $MI_{2|1}^{\boldsymbol{\pi}}(S_2'; S_1, A_1|S_2, A_2)$ that captures transition interactions between agents. Optimizing this objective encourages agent 1 to visited critical points where it can influence the transition probability of agent 2. We call such an exploration method *exploration via information-theoretic influence* (EITI).

Optimizing $MI_{2|1}^{\boldsymbol{\pi}}$ with respect to the policy parameters $\theta_1$ of agent 1 is a little bit challenging, because it is an expectation with respect to a distribution that depends on $\theta_1$. The gradient consists of two terms:

$$\nabla_{\theta_1} MI^{\boldsymbol{\pi}}(S_2'; S_1, A_1|S_2, A_2) = \sum_{\boldsymbol{s}, \boldsymbol{a}, s_2' \in (S, A, S_2)} \nabla_{\theta_1}(p^{\boldsymbol{\pi}}(\boldsymbol{s}, \boldsymbol{a}, s_2')) \log \frac{p(s_2'|\boldsymbol{s}, \boldsymbol{a})}{p^{\boldsymbol{\pi}}(s_2'|s_2, a_2)}$$
$$+ \sum_{\boldsymbol{s}, \boldsymbol{a}, s_2' \in (S, A, S_2)} p^{\boldsymbol{\pi}}(\boldsymbol{s}, \boldsymbol{a}, s_2') \nabla_{\theta_1} \log \frac{p(s_2'|\boldsymbol{s}, \boldsymbol{a})}{p^{\boldsymbol{\pi}}(s_2'|s_2, a_2)}. \tag{6}$$

While the second term is an expectation over the trajectory and can be shown to be zero (see Appendix B.1), it is unwieldy to deal with the first term because it requires the gradient of the stationary distribution, which depends on the policies and the dynamics of the environment. Fortunately, the gradient can still be estimated purely from sampled trajectories by drawing inspiration from the proof of the policy gradient theorem (Sutton et al., 2000).

The resulting policy gradient update is:

$$\nabla_{\theta_1} J_{\theta_1}(t) = \left(\hat{R}_1^t - \hat{V}_1^{\boldsymbol{\pi}}(s_t)\right) \nabla_{\theta_1} \log \pi_{\theta_1}(a_1^t|s_1^t) \tag{7}$$

where $\hat{V}_1^{\pi}(s_t)$ is an augmented value function of $\hat{R}_1^t = \sum_{t'=t}^h \hat{r}_1^{t'}$ and

$$\hat{r}_1^t = r^t + u_1^t + \beta \log \frac{p(s_2^{t+1}|s_1^t, s_2^t, a_1^t, a_2^t)}{p(s_2^{t+1}|s_2^t, a_2^t)}. \tag{8}$$

The third term, which we call *EITI reward*, is 0 when the agents are transition-independent, *i.e.*, when $p(s_2^{t+1}|s_1^t, s_2^t, a_1^t, a_2^t) = p(s_2^{t+1}|s_2^t, a_2^t)$, and is positive when $s_1^t, a_1^t$ increase the probability of agent 2 translating to $s_2^{t+1}$. Therefore, the EITI reward is an intrinsic motivation that encourages agent 1 to visit more frequently the state-action pairs where it can influence the trajectory of agent 2. The estimation of $p(s_2^{t+1}|s_1^t, s_2^t, a_1^t, a_2^t)$ and $p(s_2^{t+1}|s_2^t, a_2^t)$ are discussed in Appendix C. We assume that agents know the states and actions of other agents, but this information is only available during centralized training. When execution, agents only have access to their local observations.

## 3.2 EXPLORATION VIA DECISION-THEORETIC INFLUENCE

Mutual information characterizes the influence of one agent's trajectory on that of the other and captures interactions between the transition functions of the agents. However, it does not provide the value of these interactions to identify interactions related to more internal and external rewards ($\tilde{r}$). To address this issue, we propose *exploration via decision-theoretic influence* (EDTI) based on a decision-theoretic measure of $I$, called *Value of Interaction* (VoI), which disentangles both transition and reward influences. VoI is defined as the expected difference between the action-value function of one agent (e.g., agent 2) and its counterfactual action-value function without considering the state and action of the other agent (e.g., agent 1):

$$VoI_{2|1}^{\pi}(S_2'; S_1, A_1|S_2, A_2) = \sum_{\boldsymbol{s}, \boldsymbol{a}, s_2' \in (S, A, S_2)} p^{\pi}(\boldsymbol{s}, \boldsymbol{a}, s_2') \left[ Q_2^{\pi}(\boldsymbol{s}, \boldsymbol{a}, s_2') - Q_{2|1}^{\pi,*}(s_2, a_2, s_2') \right], \tag{9}$$

where $Q_2^{\pi}(\boldsymbol{s}, \boldsymbol{a}, s_2')$ is the expected rewards (including intrinsic rewards) of agent 2 defined as:

$$Q_2^{\pi}(\boldsymbol{s}, \boldsymbol{a}, s_2') = \tilde{r}_2(\boldsymbol{s}, \boldsymbol{a}) + \gamma \sum_{s_1'} p(s_1'|\boldsymbol{s}, \boldsymbol{a}, s_2') V_2^{\pi}(\boldsymbol{s}'), \tag{10}$$

and the counterfactual action-value function $Q_2^{\pi,*}$ (also includes intrinsic and extrinsic rewards) can be obtained by marginalizing out the state and action of agent 1:

$$Q_{2|1}^{\pi,*}(s_2, a_2, s_2') = \sum_{s_1^*, a_1^*} p^{\pi}(s_1^*, a_1^*|s_2, a_2)[\tilde{r}_2(s_1^*, s_2, a_1^*, a_2) + \gamma \sum_{s_1'} p(s_1'|s_1^*, s_2, a_1^*, a_2, s_2') V_2^{\pi}(\boldsymbol{s}')]. \tag{11}$$

Note that the definition of VoI is analogous to that of MI and the difference lies in that $\log p(\cdot)$ measures the amount of information while $Q$ measures the action value. Although VoI can be obtained by learning $Q_2^{\pi}(\boldsymbol{s}, \boldsymbol{a})$ and $Q_2^{\pi}(s_2, a_2)$ and calculating the difference, we propose to explicitly marginalize out $s_1^*$ and $a_1^*$ utilizing the estimated model transition probability $p^{\pi}(s_2'|s_2, a_2)$ and $p(s_2'|\boldsymbol{s}, \boldsymbol{a})$ to get a more accurate value estimate (Feinberg et al., 2018). The performance of these two formulations are compared in the experiments.

Value functions $Q$ and $V$ used in VoI contains both expected *external* rewards and *internal* rewards, which will not only encourage coordinated exploration by the influence between intrinsic rewards but also filter out meaningless interactions which can not lead to extrinsic reward after intrinsic reward diminishes. To facilitate the optimization of VoI, we rewrite it as an expectation over state-action trajectories.

$$VoI_{2|1}^{\pi}(S_2'; S_1, A_1|S_2, A_2) = \mathbb{E}_{\tau} \left[ \tilde{r}_2(\boldsymbol{s}, \boldsymbol{a}) - \tilde{r}_2^{\pi}(s_2, a_2) + \gamma \left( 1 - \frac{p^{\pi}(s_2'|s_2, a_2)}{p(s_2'|\boldsymbol{s}, \boldsymbol{a})} \right) V_2^{\pi}(\boldsymbol{s}') \right], \tag{12}$$

where $\tilde{r}_2^{\pi}(s_2, a_2)$ is the counterfactual immediate reward. The detailed proof is deferred to Appendix B.2. From this definition, we can intuitively see how VoI reflects the value of interactions. $\tilde{r}_2(\boldsymbol{s}, \boldsymbol{a}) - \tilde{r}_2^{\pi}(s_2, a_2)$ and $1 - p^{\pi}(s_2'|s_2, a_2)/p(s_2'|\boldsymbol{s}, \boldsymbol{a})$ measure the influence of agent 1 on the immediate reward and the transition function of agent 2, and $V_2^{\pi}(\boldsymbol{s}')$ serves as a scale factor in terms of future value. Only when agent 1 and agent 2 are both transition- and reward-independent, *i.e.*, when $p^{\pi}(s_2'|s_2, a_2) = p(s_2'|\boldsymbol{s}, \boldsymbol{a})$ and $r_2^{\pi}(s_2, a_2) = r_2(\boldsymbol{s}, \boldsymbol{a})$ will VoI equal to 0. In particular, maximizing

VoI with respect to policy parameters $\theta_1$ will lead agent 1 to meaningful interaction points, where $V_2^{\boldsymbol{\pi}}(\boldsymbol{s}')$ is high and $s_1, a_1$ can increase the probability that $\boldsymbol{s}'$ is reached.

In this learning framework, agents initially explore the environment individually driven by its own curiosity, during which process they will discover potentially valuable interaction points where they can influence the transition function and (intrinsic) rewarding structure of each other. VoI highlights these points and encourages agents to visit these configurations more frequently. As intrinsic reward diminishes, VoI can gradually distinguish those interaction points which are necessary to get extrinsic rewards.

### 3.2.1 Policy Optimization with VoI

We want to optimize $J_{\theta_i}$ with respect to the policy parameters $\theta_i$, where the most cumbrous term is $\nabla_{\theta_i} VoI_{-i|i}$. For brevity, we can consider a two-agent case, e.g., optimizing $VoI_{2|1}$ with respect to the policy parameters $\theta_1$. Directly computing the gradient $\nabla_{\theta_1} VoI_{2|1}$ is not stable, because $VoI_{2|1}$ contains policy-dependent functions $\tilde{r}_2^{\boldsymbol{\pi}}(s_2, a_2)$, $p^{\boldsymbol{\pi}}(s_2'|s_2, a_2)$, and $V_2^{\boldsymbol{\pi}}(\boldsymbol{s}')$ (see Eq. 12). To stabilize training , we use target functions to approximate these policy-dependent functions, which is a commonly used technique in deep RL (Mnih et al., 2015). With this approximation, we denote

$$g_2(\boldsymbol{s}, \boldsymbol{a}) = \tilde{r}_2(\boldsymbol{s}, \boldsymbol{a}) - \tilde{r}_2^-(s_2, a_2) + \gamma \sum_{\boldsymbol{s}'} T(\boldsymbol{s}'|\boldsymbol{s}, \boldsymbol{a}) \left(1 - \frac{p^-(s_2'|s_2, a_2)}{p(s_2'|\boldsymbol{s}, \boldsymbol{a})}\right) V_2^-(s_1', s_2'). \quad (13)$$

where $r_2^-$, $p^-$, and $V_2^-$ are corresponding target functions. As these target functions are only periodically updated during the learning, their gradients over $\theta_1$ can be approximately ignored. Therefore, from Eq. 12, we have

$$\nabla_{\theta_1} VoI_{2|1}^{\boldsymbol{\pi}}(S_2'; S_1, A_1|S_2, A_2) \approx \sum_{\boldsymbol{s}, \boldsymbol{a} \in (S, A)} \left(\nabla_{\theta_1} p^{\boldsymbol{\pi}}(\boldsymbol{s}, \boldsymbol{a})\right) g_2(\boldsymbol{s}, \boldsymbol{a}). \quad (14)$$

Similar to the calculation of $\nabla_{\theta_i} MI$, we get the gradient at every step (see Appendix B.3 for proof):

$$\nabla_{\theta_1} J_{\theta_1}(t) \approx \left(\hat{R}_1^t - \hat{V}_1^{\boldsymbol{\pi}}(s_t)\right) \nabla_{\theta_1} \log \pi_{\theta_1}(a_1^t|s_1^t), \quad (15)$$

where $\hat{V}_1^{\boldsymbol{\pi}}(s_t)$ is an augmented value function regressed towards $\hat{R}_1^t = \sum_{t'=t}^h \hat{r}_1^{t'}$ and

$$\hat{r}_1^t = r^t + u_1^t + \beta \left[u_2^t + \gamma \left(1 - \frac{p^-(s_2^{t+1}|s_2^t, a_2^t)}{p(s_2^{t+1}|s_1^t, s_2^t, a_1^t, a_2^t)}\right) V_2^-(s_1^{t+1}, s_2^{t+1})\right]. \quad (16)$$

We call $u_2^t + \gamma \left(1 - \frac{p^-(s_2^{t+1}|s_2^t, a_2^t)}{p(s_2^{t+1}|s_1^t, s_2^t, a_1^t, a_2^t)}\right) V_2^-(s_1^{t+1}, s_2^{t+1})$ the *EDTI reward*.

### 3.3 Discussions

**Scale to Large Settings:** For cases with more than two agents, the VoI of agent $i$ on other agents can be defined similarly to Eq. 9, which is annotated with $VoI_{-i|i}^{\boldsymbol{\pi}}(S_{-i}'; S_i, A_i|S_{-i}, A_{-i})$, where $S_{-i}$ and $A_{-i}$ are the state and action sets of all agents other than agent $i$. In practice, agents interaction can often be decomposed to pairwise interaction so $VoI_{-i|i}^{\boldsymbol{\pi}}(S_{-i}'; S_i, A_i|S_{-i}, A_{-i})$ is well approximated by the sum of values of pairwise value of interaction:

$$VoI_{-i|i}^{\boldsymbol{\pi}}(S_{-i}'; S_i, A_i|S_{-i}, A_{-i}) \approx \sum_{j \in N, j \neq i} VoI_{j|i}^{\boldsymbol{\pi}}(S_j'; S_i, A_i|S_{-i}, A_{-i}). \quad (17)$$

**Relationship between EITI and EDTI:** EITI and EDTI gradient updates are obtained by information- and decision-theoretical influence respectively. Therefore, it is nontrivial to derive that part of the EDTI reward is a lower bound of the EITI reward:

$$1 - \frac{p(s_{-i}'|s_{-i}, a_{-i})}{p(s_{-i}'|\boldsymbol{s}, \boldsymbol{a})} \leq \log \frac{p(s_{-i}'|\boldsymbol{s}, \boldsymbol{a})}{p(s_{-i}'|s_{-i}, a_{-i})}, \quad \forall \boldsymbol{s}, \boldsymbol{a}, s_{-i}' \quad (18)$$

which easily follows given that $\log x \geq 1 - 1/x$ for $\forall x > 0$. This draws a connection between EITI and EDTI reward.

Table 1: Baseline algorithms. The third column is the reward used to train the value function of PPO. $u_i$ and $u_{cen}$ are curiosity about individual state $s_i$ and global state $\boldsymbol{s}$, $T_1 = \log\left(p(s'_{-i}|\boldsymbol{s},\boldsymbol{a})/p(s'_{-i}|s_{-i},a_{-i})\right)$, $T_2 = 1 - p(s'_{-i}|s_{-i},a_{-i})/p(s'_{-i}|\boldsymbol{s},\boldsymbol{a})$, and $\Delta Q_{-i}(\boldsymbol{s},\boldsymbol{a}) = Q_{-i}(\boldsymbol{s},\boldsymbol{a}) - Q_{-i}(s_{-i},a_{-i})$. Social influence (Jaques et al., 2018) and COMA (Foerster et al., 2018) are augmented with curiosity.

|  | Alg. | Reward | Description |
|---|---|---|---|
| Ours | EITI | $r + u_i + \beta T_1$ | Influence-theoretic influence |
|  | EDTI | $r + u_i + \beta(u_{-i} + \gamma T_2 V_{-i})$ | Decision-theoretic influence |
| Other Exploration Methods | random | $r$ | Pure PPO |
|  | cen | $r + u_{cen}$ | Decentralized PPO with cen curiosity |
|  | dec | $r + u_i$ | Decentralized PPO with dec curiosity |
|  | cen_control | $r + u_{cen}$ | Centralized PPO with cen curiosity |
| Ablations | r_influence | $r + u_i + \beta u_{-i}$ | Disentangle reward interaction |
|  | plusV | $r + u_i + \beta V_{-i}$ | Use other agents' value functions |
|  | shared_critic | $r + u_{cen}$ | PPO with shared $V$ and cen curiosity |
|  | Q-Q | $r + u_i + \beta \Delta Q_{-i}(\boldsymbol{s},\boldsymbol{a})$ | EDTI without explicit counterfactual |
| Related Works | social | — | By Jaques et al. (2018) |
|  | COMA | — | By Foerster et al. (2018) |
|  | Multi | — | By Iqbal & Sha (2019b) |

**Comparing EDTI to Centralized Methods:** Different from a centralized method which directly includes value functions of other agents in the optimization objective, (*i.e.*, by setting total reward $\hat{r}_i = r + u_i + \beta(u_{-i} + \gamma V_{-i})$, which is called *plusV* henceforth), the EDTI reward for agent $i$ disentangles its contributions to values of another agents using a counterfactual formulation. This difference is important for quantifying influence because the value of another agent does not just contain the contributions from agent $i$, but also those of itself and third-party agents. Therefore, EDTI is a kind of *intrinsic reward assignment*. Our experiments in the next section will compare the performance of *plusV* against our methods, which verify the importance of the intrinsic reward assignment.

## 4 EXPERIMENTAL RESULTS

Our experiments aim to answer the following questions: (1) Can EITI and EDTI rewards capture interaction points? If they can, how do these points change throughout exploration? (2) Can exploiting these interaction points facilitate exploration and learning performance? (3) Can EDTI filter out interaction points that are not related to environmental rewards? (4) What if only reward influence between agents are disentangled? We evaluate our approach on a set of multi-agent tasks with sparse rewards based on a discrete version of multi-agent particle world environment (Lowe et al., 2017). PPO (Schulman et al., 2017) is used as the underlying algorithm. For evaluation, all experiments are carried out with 5 different random seeds and results are shown with $95\%$ confidence interval. Demonstrative videos[1] are available online.

**Baselines** We compare our methods with various baselines shown in Table 1. In particular, we carry out the following ablation studies: i) r_influence disentangles immediate reward influence between agents, (derivation of the associated augmented reward can be found in Appendix B.4. Reward influence in long term is not considered because it inevitably involves transition interactions) ii) PlusV as described in Sec. 3.3. iii) Shared_critic uses decentralized PPO agents with shared centralized value function and thus is a cooperative version of MADDPG (Lowe et al., 2017) augmented with intrinsic reward of curiosity. iv) Q-Q is similar to EDTI but without explicit counterfactual formulation, as described in Sec. 3.2. We also note that EITI is an ablation of EDTI which considers transition interactions. PlusV, shared_critic, Q-Q, and cen_control have access to global or other agents' value functions during training. When execution, all the methods except cen_control only require local state.

---

[1]https://sites.google.com/view/influence-based-ma-exploration/

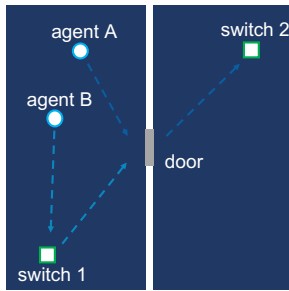 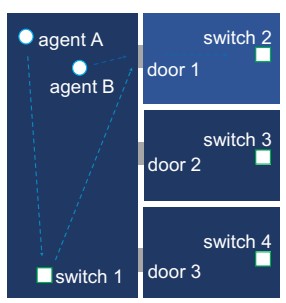 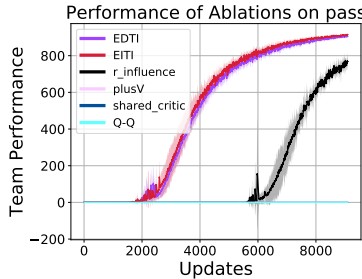

Figure 1: Didactic examples. Left: task **Pass**. Two agents starting at the upper-left corner are only rewarded when both of them reach the other room through the door, which will open only when at least one of the switches is occupied by one or more agents. Middle: **Secret-Room**. An extension of *Pass* with 4 rooms and switches. When the switch 1 is occupied, all the three doors turn open. And the three switches on the right only control the door of its room. The agents need to reach the upper right room to achieve any reward. Right: comparison of our methods with ablations on *Pass*.

## 4.1 DIDACTIC EXAMPLES

We present two didactic examples of multi-agent cooperation tasks with sparse reward to explain how EITI and EDTI work. The first didactic example consists of a $30 \times 30$ maze with two rooms and a door with two switches (Fig. 1 left). In the optimal strategy, one agent should first step on switch 1 to help the other agent pass the door, and then the agent that has already reached the right half should further go to switch 2 to bring the remaining agent in. There are two pairs of interaction points in this task: (switch 1, door) and (switch 2, door), *i.e.*, transition probability of the agent near door is determined by whether another agent is on one of the switch.

Fig. 1-right and Fig. 2-top show the learning curves of our methods and all the baselines, among which EITI, EDTI, r_influence, Multi, and centralized control can learn the winning strategy and ours learn much more efficiently. Fig. 2-bottom gives a possible explanation why our methods work. EITI and EDTI rewards successfully highlight the interaction points (before 100 and 2100 updates, respectively). Agents are encouraged to explore these configurations more frequently and thus have better chance to learn the goal strategy. EDTI reward considers the value function of the other agent, so it converges slower than the EITI reward. In contrast, directly adding the other agent's intrinsic rewards and value functions is noisy (see "plusV reward") and confuses the agent because these contain the effect of the other agent's exploration. As for centralized control, global curiosity encourages agents to try all possible configurations, so it can find environmental rewards in most tasks. However, visiting all configurations without bias renders it inefficient – external rewards begin to dominate the behaviors of agents after 7000 updates even with the help of centralized learning algorithm. Our methods use the same information as centralized exploration but take advantages of agents' interactions to accelerate exploration.

In order to evaluate whether EDTI can help filter out noisy interaction points and accelerate exploration, we conduct experiments in a second didactic task (see Fig. 1 middle). It is also a navigation task in a $25 \times 25$ maze where agents are rewarded for being in a goal room. However, in this experiment, we consider a case where there are four rooms and the upper right one is attached to reward. This task contains 6 pairs of interaction points (switch 1 with each of the doors, each switch with the door of the same room), but only two of them are related to external rewards, *i.e.*, (switch 1, door 1) and (switch 2, door 1). As Fig. 3-right shows, EITI agents treat three doors equally even after 7400 updates (see Fig. 3 right, 7400 updates, top row). In comparison, although EDTI reward suffers from noise in the beginning, it clearly highlight two pairs of valuable interaction points (see Fig. 3 right, 7400 updates, bottom row) as intrinsic reward diminishes. This can explain why EDTI outperforms EITI (Fig. 3 left).

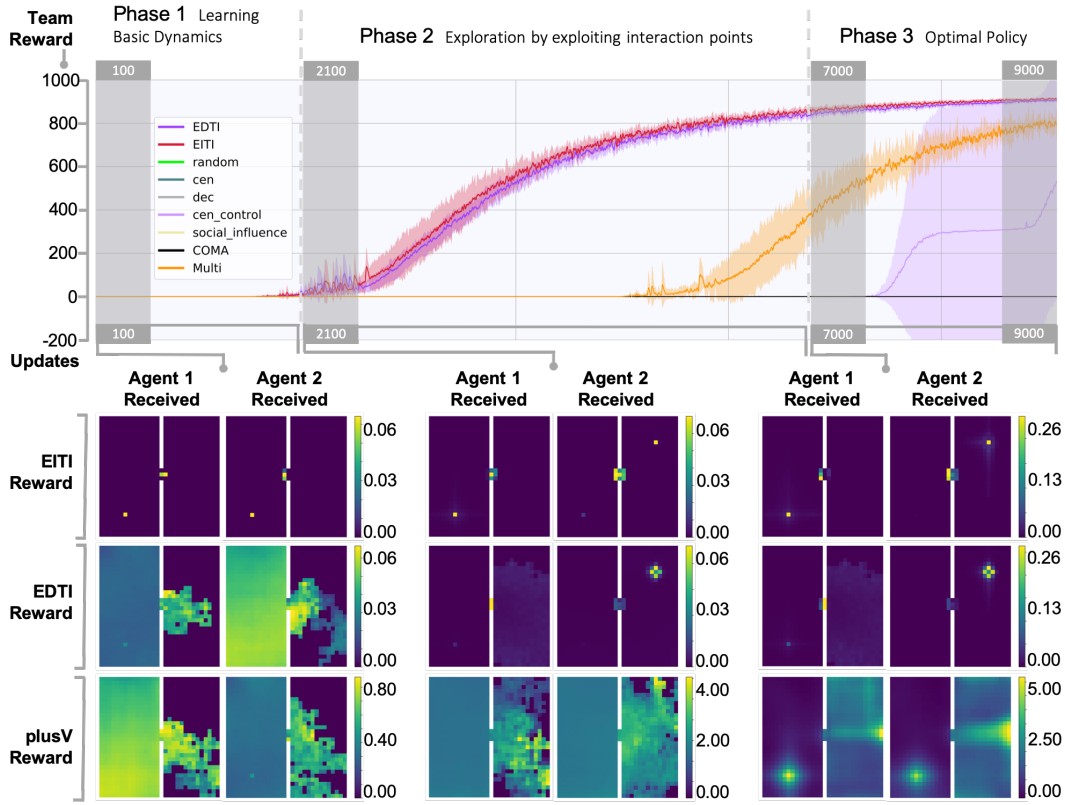

Figure 2: Development of performance of our methods compared to baselines and intrinsic reward terms of EITI, EDTI, and plusV over the training period of 9000 PPO updates segmented into three phases. "Team Reward" shows averaged team reward gained in a episode, with a maximum of 1000. It shows that only EITI, EDTI, and centralized control and Multi can learn the strategy during this stage. "EITI reward", "EDTI reward", and "plusV reward" demonstrate the evolving of corresponding intrinsic rewards.

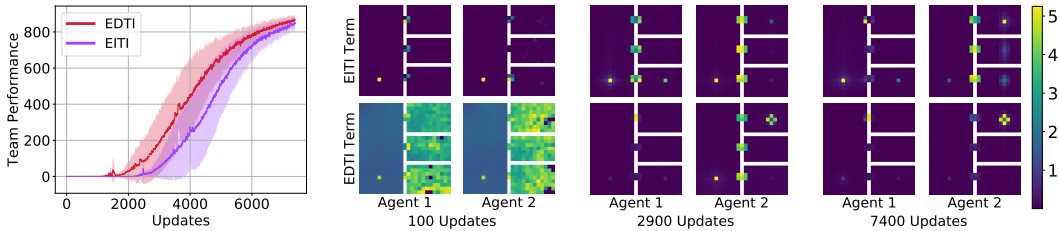

Figure 3: Left: performance comparison between EDTI and EITI on *Secret-Room* over 7400 PPO updates. Right: EITI and EDTI terms of two agents after 100, 2900, and 7400 updates.

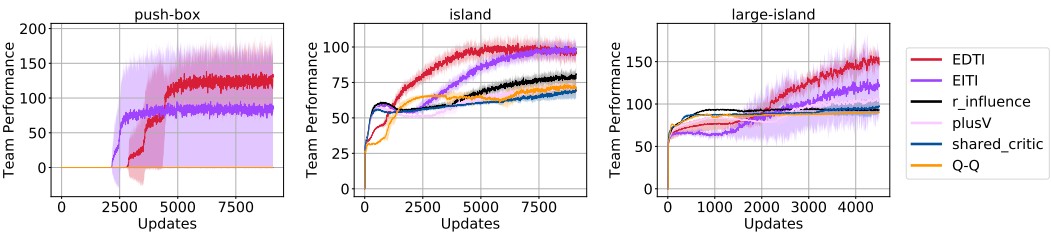

Figure 4: Comparison of our methods against ablations for *Push-Box*, *Island*, and *Large-Island*. Comparison with baselines is shown in Fig. 8 in Appendix D.

## 4.2 Exploration in Complex Tasks

Next, we evaluate the performance of our methods on more complex tasks. To this end, we use three sparse reward cooperative multi-agent tasks depicted in Fig. 7 of Appendix D and analyzed below. Details of implementation and experiment settings are also described in Appendix D.

**Push-Box:** A $15 \times 15$ room is populated with 2 agents and 1 box. Agents need to push the box to the wall in 300 environment steps to get a reward of 1000. However, the box is so heavy that only when two agents push it in the same direction at the same time can it be moved a grid. Agents need to coordinate their positions and actions for multiple steps to earn a reward. The purpose of this task is to demonstrate that EITI and EDTI can explore long-term cooperative strategy.

**Island:** This task is a modified version of the classic Stag Hunt game (Peysakhovich & Lerer, 2018) where two agents roam a $10 \times 10$ island populated with 9 treasures and a random walking beast for 300 environment steps. Agents can collect a treasure by stepping on it to get a team reward of 10 or, by attacking the beast within their attack range, capture it for a reward of 300. The beast would also attack the agents when they are too close. The beast and agent have a maximum energy of 8 and 5 respectively, which will be subtracted by 1 every time attacked. Therefore, an agent is too weak to beat the beast alone and they have to cooperate. In order to learn optimal strategy in this task, one method has to keep exploring after sub-optimal external rewards are found.

**Large-Island:** Similar to *Island* but with more agents (4), more treasures (16), and a beast with more energy (16) and a higher reward (600) for being caught. This task aims to demonstrate feasibility of our methods in cases with more than 2 agents.

*Push-Box* requires agents to take coordinated actions at certain positions for multiple steps to get rewarded. Therefore, this task is particularly challenging and all the baselines struggle to earn any reward (Fig. 4 left and Fig. 8 left). Our methods are considerably more successful because interaction happens when the box is moved – agents remain unmoved when they push the box alone but will move by a grid if push it together. In this way, EITI and EDTI agents are rewarded intrinsically to move the box and thus are able to quickly find the optimal policy.

In the *Island* task, collecting treasures is a easily-attainable local optimal. However, efficient treasures collecting requires the agents to spread on the island. This leads to a situation where attempting to attack the beast seems a bad choice since it is highly possible that agents will be exposed to the beast's attack alone. They have to give up profitable spreading strategy and take the risk of being killed to discover that if they attack the beast collectively for several timesteps, they will get much more rewards. Our methods help solve this challenge by giving agents intrinsic incentives to appear together in the attack range of the beast, where they have indirect interactions (health is part of the state and it decreases slower when the two are attacked alternatively). Fig. 9 in Appendix D demonstrates that our methods learn to catch the beast quickly, and thus have better performance (Fig. 8 right).

Finally, outperformance of our methods on *Large-Island* proves that they can successfully handle cases with more than two agents.

In summary, both of our methods are able to facilitate effective exploration on all the tasks by exploiting interactions. EITI outperforms EDTI in scenarios where all interaction points align with extrinsic rewards. On other tasks, EDTI performs better than EITI due to its ability to filter out interaction points that can not lead to more values.

We also study EDTI with only intrinsic rewards, discussion and results are included in Appendix A.

## 5 Related Works

Single-agent exploration achieves conspicuous success recently. Provably efficient methods are proposed, such as upper confidence bound (UCB) (Jaksch et al., 2010; Azar et al., 2017; Jin et al., 2018) and posterior sampling for reinforcement learning (PSRL) (Strens, 2000; Osband et al., 2013; Osband & Van Roy, 2016; Agrawal & Jia, 2017). Given that these methods do not scale well to large or continuous settings, another line of research has been focusing on curiosity-driven exploration (Schmidhuber, 1991; Chentanez et al., 2005; Oudeyer et al., 2007; Barto, 2013; Bellemare et al., 2016; Pathak et al., 2017; Ostrovski et al., 2017), and have shown impressive results (Burda

et al., 2019; 2018; Hyoungseok Kim, 2019). In addition, methods based on variational information maximization (Houthooft et al., 2016; Barron et al., 2018) and mutual information (Rubin et al., 2012; Still & Precup, 2012; Salge et al., 2014; Mohamed & Rezende, 2015; Hyoungseok Kim, 2019) have been proposed for single-agent intrinsically motivated exploration.

Although multi-agent reinforcement learning (MARL) has been making significant progresses in recent years (Foerster et al., 2018; Lowe et al., 2017; Wen et al., 2019; Iqbal & Sha, 2019a; Sunehag et al., 2018; Son et al., 2019; Rashid et al., 2018), less attention has been drawn to multi-agent exploration. Dimakopoulou & Van Roy (2018) and Dimakopoulou et al. (2018) propose posterior sampling methods for exploration of concurrent reinforcement learning in coverage problems, Bargiacchi et al. (2018) presents a multi-agent upper confidence exploration method for repeated single-stage problems, and Iqbal & Sha (2019b) investigates methods to combine several decentralized curiosity-driven exploration strategies. All these works focus on transition-independent settings. Another Bayesian exploration approach has been proposed for learning in stateless repeated games (Chalkiadakis & Boutilier, 2003). In contrast, this paper focuses on more general multi-agent sequential decision making problems with complex reward dependencies and transition interactions among agents.

In the literature of MARL, COMA (Foerster et al., 2018) shares some similarity with our decision-theoretic EDTI approach in that both of them use the idea of counterfactual formulations. However, they are quite different in terms of definition and optimization: (1) conceptually, EDTI measures the influence of one agent on the value functions of other agents, while COMA quantifies individual contribution to the team value; (2) EDTI is defined on counterfactual Q-value over state-action pairs of other agents given its own state-action pair, while COMA uses the counterfactual Q-value just over its own action without considering state information, which is critical for exploration; (3) we explicitly derive the gradients for optimizing EDTI influence for coordinated exploration in the policy gradient framework, which provides more accurate feedback, while COMA uses the counterfactual Q value as a critic. Another line of relevant works (Oliehoek et al., 2012; de Castro et al., 2019) propose influence-based abstraction to predict influence sources to help local decision making of agents. In contrast, this paper presents two novel approaches that quantify and maximize the influence between agents for enabling coordinated multi-agent exploration.

In addition, some previous MARL work has also studied intrinsic rewards. One notably relevant work is Jaques et al. (2018), which models the social influence of one agent on other agents' policies. In contrast, EITI measures the influence of one agent on the transition dynamics of other agents. Accompanying this distinction, EITI includes states of agents in the calculation of influence while social influence dos not. Apart from that, the optimization methods are also different – we directly derive the gradients of mutual information and incorporate its optimization in the policy gradient framework, while Jaques et al. (2018) adds social influence reward to the immediate environmental reward for training policies. Hughes et al. (2018) proposes an inequality aversion reward for learning in intertemporal social dilemmas. Strouse et al. (2018) uses mutual information between goal and states or actions as an intrinsic reward to train the agent to share or hide their intentions.

## 6 Closing Remarks

In this paper, we study the multi-agent exploration problem and propose two influence-based methods that exploits the interaction structure. These methods are based on two interaction measures, MI and *Value of Interaction* (VoI), which respectively measure the amount and value of one agent's influence on the other agents' exploration processes. These two measures can be best regraded as exploration bonus distribution. We also propose an optimization method in the policy gradient framework, which enables agents to achieve coordinated exploration in a decentralized manner and optimize team performance.

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

## APPENDIX

## A   INTRINSIC EDTI

Value of interaction (VoI) captures both transition and reward influence among agents, and it facilitates coordinated exploration by encouraging interactions. VoI contains influence of both intrinsic and extrinsic rewards. Since single-agent literature has studied purely curiosity-driven learning and gets cutting-edge performance (Burda et al., 2019), it is interesting to investigate the performance of VoI given only intrinsic rewards.

Intuitively, intrinsic VoI distributes individual curiosity among team members and facilitates exploration by encouraging agents to help each other to reach under-explored states. Specifically, we use the following objective:

$$J_{\theta_i}[\pi_i|\pi_{-i}, p_0] \equiv V^{ext,\boldsymbol{\pi}}(\boldsymbol{s}_0) + V_i^{int,\boldsymbol{\pi}}(\boldsymbol{s}_0) + \beta \cdot VoI_{-i|i}^{int,\boldsymbol{\pi}}. \tag{19}$$

The corresponding augmented reward is:

$$\hat{r}_1^t = r_t + u_1^t + \beta \left[ u_2^t + \gamma \left( 1 - \frac{p^-(s_2^{t+1}|s_2^t, a_2^t)}{p(s_2^{t+1}|s_1^t, s_2^t, a_1^t, a_2^t)} \right) V_2^{int,-}(s_1^{t+1}, s_2^{t+1}) \right] \tag{20}$$

We use this method (intrinsic EDTI) to train the agents on *Pass*, *Secret-Room*, *Push-Box*, and *Island* and show the results in Fig. 5.

## B   MATHEMATICAL DETAILS

### B.1   GRADIENT OF MUTUAL INFORMATION

To encourage agents to exert influence on transitions of other agents, we optimize mutual information between agent's trajectories. In particular, in the following, we show that term 2 in Eq. 6 is always zero.

$$T2 = \sum_{\boldsymbol{s},\boldsymbol{a},s_2' \in (S,A,S_2)} p^{\boldsymbol{\pi}}(\boldsymbol{s},\boldsymbol{a},s_2') \nabla_{\theta_1} \log \frac{p(s_2'|\boldsymbol{s},\boldsymbol{a})}{p^{\boldsymbol{\pi}}(s_2'|s_2,a_2)} \tag{21}$$

$$= -\sum_{\boldsymbol{s},\boldsymbol{a},s_2'} p^{\boldsymbol{\pi}}(\boldsymbol{s},\boldsymbol{a},s_2') \nabla_{\theta_1} \log p^{\boldsymbol{\pi}}(s_2'|s_2,a_2) \tag{22}$$

$$= -\sum_{\boldsymbol{s},\boldsymbol{a},s_2'} p^{\boldsymbol{\pi}}(\boldsymbol{s},\boldsymbol{a},s_2') \frac{\nabla_{\theta_1}(p^{\boldsymbol{\pi}}(s_2'|s_2,a_2))}{p^{\boldsymbol{\pi}}(s_2'|s_2,a_2)} \tag{23}$$

$$= -\sum_{\boldsymbol{s},\boldsymbol{a},s_2'} \frac{p^{\boldsymbol{\pi}}(\boldsymbol{s},\boldsymbol{a},s_2')}{p^{\boldsymbol{\pi}}(s_2'|s_2,a_2)} \nabla_{\theta_1} \left( \sum_{s_1^*,a_1^*} p(s_2'|s_2,a_2,s_1^*,a_1^*)p(s_1^*|s_2,a_2)\pi_{\theta_1}(a_1^*|s_1^*) \right) \tag{24}$$

$$= -\sum_{\boldsymbol{s},\boldsymbol{a},s_2'} \frac{p^{\boldsymbol{\pi}}(\boldsymbol{s},\boldsymbol{a},s_2')}{p^{\boldsymbol{\pi}}(s_2'|s_2,a_2)} \sum_{s_1^*,a_1^*} p(s_2'|s_2,a_2,s_1^*,a_1^*)p(s_1^*|s_2,a_2)\nabla_{\theta_1}\pi_{\theta_1}(a_1^*|s_1^*) \tag{25}$$

$$= -\sum_{s_2,a_2,s_2'} \frac{p^{\boldsymbol{\pi}}(s_2,a_2,s_2')}{p^{\boldsymbol{\pi}}(s_2'|s_2,a_2)} \sum_{s_1^*,a_1^*} p(s_2'|s_2,a_2,s_1^*,a_1^*)p(s_1^*|s_2,a_2)\nabla_{\theta_1}\pi_{\theta_1}(a_1^*|s_1^*) \tag{26}$$

$$= -\sum_{s_2,a_2,s_2'} p^{\boldsymbol{\pi}}(s_2,a_2) \sum_{s_1^*,a_1^*} p(s_2'|s_2,a_2,s_1^*,a_1^*)p(s_1^*|s_2,a_2)\nabla_{\theta_1}\pi_{\theta_1}(a_1^*|s_1^*) \tag{27}$$

$$= -\sum_{s_2,a_2} p^{\boldsymbol{\pi}}(s_2,a_2) \sum_{s_1^*,a_1^*} p(s_1^*|s_2,a_2)\nabla_{\theta_1}\pi_{\theta_1}(a_1^*|s_1^*) \sum_{s_2'} p(s_2'|s_2,a_2,s_1^*,a_1^*) \tag{28}$$

$$= -\sum_{s_2,a_2} p^{\boldsymbol{\pi}}(s_2,a_2) \sum_{s_1^*,a_1^*} p(s_1^*|s_2,a_2)\nabla_{\theta_1}\pi_{\theta_1}(a_1^*|s_1^*) \underbrace{\sum_{s_2'} p(s_2'|s_2,a_2,s_1^*,a_1^*)}_{=1} \tag{29}$$

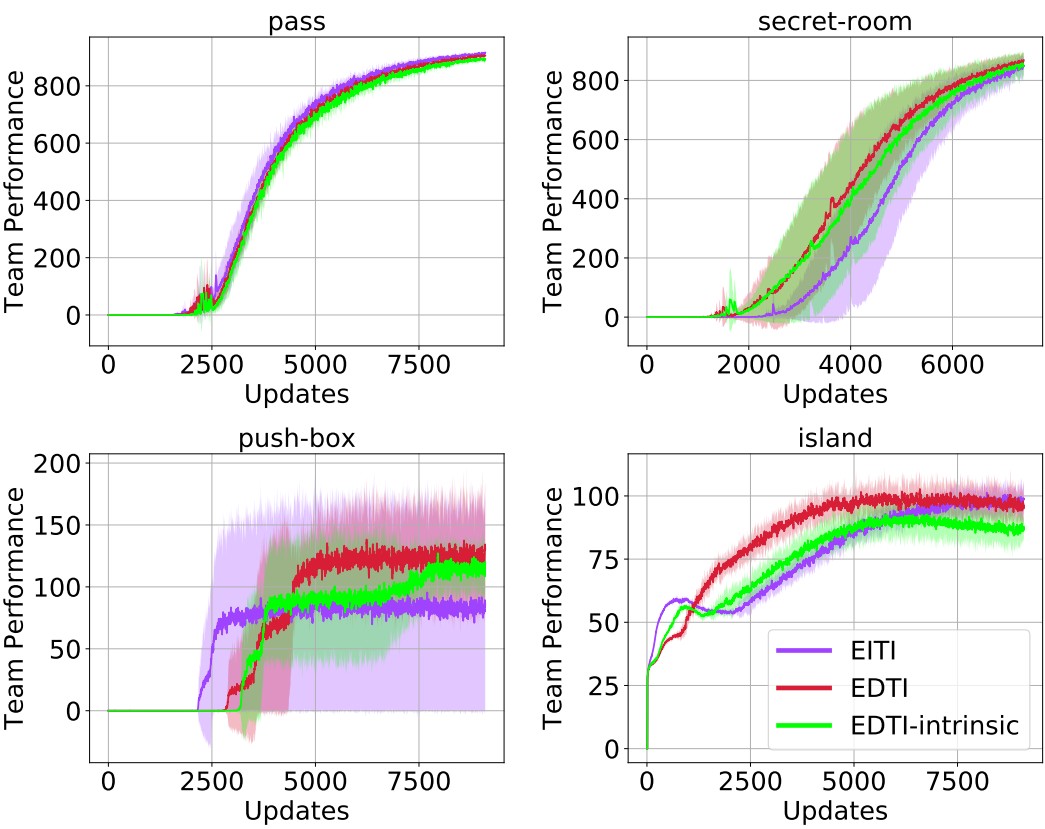

Figure 5: Performance of intrinsic EDTI in comparison with EITI and EDTI on *Pass*, *Secret-Room*, *Push-Box*, and *Island*.

$$
= -\sum_{s_2,a_2} p^{\boldsymbol{\pi}}(s_2,a_2) \sum_{s_1^*} p(s_1^*|s_2,a_2)\nabla_{\theta_1} \sum_{a_1^*} \pi_{\theta_1}(a_1^*|s_1^*) \tag{30}
$$

$$
= -\sum_{s_2,a_2} p^{\boldsymbol{\pi}}(s_2,a_2) \sum_{s_1^*} p(s_1^*|s_2,a_2)\nabla_{\theta_1} 1 \tag{31}
$$

$$
= 0 \tag{32}
$$

### B.2   DEFINITION OF *Value of Interaction*

To capture both transition and reward interactions between agents and thereby achieve intrinsic reward distribution, we propose a decision-theoretic measure called *Value of Interaction*. We start from 2-agent cases and the following theorem gives the definition of $VoI_{2|1}$ in the form of an expectation over trajectories, which is especially helpful in the derivation of the EDTI policy gradient update shown Eq. 15.

**Theorem 1.** *Value of Interaction of agent 1 on agent 2 is:*

$$
VoI_{2|1}^{\boldsymbol{\pi}}(S_2'; S_1, A_1|S_2, A_2) = \mathbb{E}_{\tau}\left[\tilde{r}_2(\boldsymbol{s},\boldsymbol{a}) - \tilde{r}_2^{\boldsymbol{\pi}}(s_2,a_2) + \gamma\left(1 - \frac{p^{\boldsymbol{\pi}}(s_2'|s_2,a_2)}{p(s_2'|\boldsymbol{s},\boldsymbol{a})}\right)V_2^{\boldsymbol{\pi}}(\boldsymbol{s}')\right], \tag{33}
$$

*where $\tilde{r}_2^{\boldsymbol{\pi}}(s_2,a_2)$ is the counterfactual immediate reward.*

$VoI_{2|1}$ can be defined similarly. To lighten notation in the proof, we define

$$
V_2^{\boldsymbol{\pi}}(s_2'|s_1,s_2,a_1,a_2) = \sum_{s_1'} p(s_1'|s_1,s_2,a_1,a_2,s_2')V_2^{\boldsymbol{\pi}}(s_1',s_2') \tag{34}
$$

$$\tilde{r}_2^{\boldsymbol{\pi}}(s_2, a_2) = \sum_{s_1^*, a_1^*} p^{\boldsymbol{\pi}}(s_1^*, a_1^* | s_2, a_2) \tilde{r}_2(s_1^*, s_2, a_1^*, a_2), \tag{35}$$

$$V_2^{\boldsymbol{\pi}, *}(s_2' | s_2, a_2) = \sum_{s_1^*, a_1^*} p^{\boldsymbol{\pi}}(s_1^*, a_1^* | s_2, a_2) \sum_{s_1'} p(s_1' | s_1^*, s_2, a_1^*, a_2, s_2') V_2^{\boldsymbol{\pi}}(s_1', s_2'). \tag{36}$$

We first prove Lemma 1, which is used in the proof of Theorem 1.

**Lemma 1.**

$$\sum_{s_1, s_2, a_1, a_2} p^{\boldsymbol{\pi}}(s_1, s_2, a_1, a_2) \gamma \sum_{s_2'} p(s_2' | s_1, s_2, a_1, a_2) V_2^{\boldsymbol{\pi}}(s_2' | s_2, a_2) \tag{37}$$

$$= \sum_{s_1, s_2, a_1, a_2} p^{\boldsymbol{\pi}}(s_1, s_2, a_1, a_2) \gamma \sum_{s_1', s_2'} T(s_1', s_2' | s_1, s_2, a_1, a_2) \cdot \frac{p^{\boldsymbol{\pi}}(s_2' | s_2, a_2)}{p(s_2' | s_1, s_2, a_1, a_2)} V_2^{\boldsymbol{\pi}}(s_1', s_2').$$

*Proof.*

$$\sum_{s_1, s_2, a_1, a_2} p^{\boldsymbol{\pi}}(s_1, s_2, a_1, a_2) \gamma \sum_{s_2'} p(s_2' | s_1, s_2, a_1, a_2) V_2^{\boldsymbol{\pi}}(s_2' | s_2, a_2) \tag{38}$$

$$= \sum_{s_1, s_2, a_1, a_2} p^{\boldsymbol{\pi}}(s_1, s_2, a_1, a_2) \gamma \sum_{s_2'} p(s_2' | s_1, s_2, a_1, a_2) \cdot \tag{39}$$

$$\sum_{s_1^*, a_1^*} p^{\boldsymbol{\pi}}(s_1^*, a_1^* | s_2, a_2) \sum_{s_1'} p(s_1' | s_1^*, s_2, a_1^*, a_2, s_2') V_2^{\boldsymbol{\pi}}(s_1', s_2') \tag{40}$$

$$= \sum_{s_1, s_2, a_1, a_2} p^{\boldsymbol{\pi}}(s_1, s_2, a_1, a_2) \gamma \sum_{s_2'} p(s_2' | s_1, s_2, a_1, a_2) \cdot \tag{41}$$

$$\sum_{s_1^*, a_1^*} p^{\boldsymbol{\pi}}(s_1^*, a_1^* | s_2, a_2) \sum_{s_1'} \frac{T(s_1', s_2' | s_1^*, s_2, a_1^*, a_2)}{p(s_2' | s_1^*, s_2, a_1^*, a_2)} V_2^{\boldsymbol{\pi}}(s_1', s_2') \tag{42}$$

$$= \sum_{s_1, s_2, a_1, a_2} p^{\boldsymbol{\pi}}(s_1, s_2, a_1, a_2) \gamma \sum_{s_1', s_2'} \frac{T(s_1', s_2' | s_1, s_2, a_1, a_2)}{p(s_2' | s_1, s_2, a_1, a_2)} \cdot \tag{43}$$

$$V_2^{\boldsymbol{\pi}}(s_1', s_2') \sum_{s_1^*, a_1^*} p^{\boldsymbol{\pi}}(s_1^*, a_1^* | s_2, a_2) p(s_2' | s_1^*, s_2, a_1^*, a_2) \tag{44}$$

$$= \sum_{s_1, s_2, a_1, a_2} p^{\boldsymbol{\pi}}(s_1, s_2, a_1, a_2) \gamma \sum_{s_1', s_2'} T(s_1', s_2' | s_1, s_2, a_1, a_2) \cdot \tag{45}$$

$$\frac{p^{\boldsymbol{\pi}}(s_2' | s_2, a_2)}{p(s_2' | s_1, s_2, a_1, a_2)} V_2^{\boldsymbol{\pi}}(s_1', s_2'). \tag{46}$$

$$\square$$

We now give the proof of Theorem 1:

*Proof.*

$$VoI_{2|1}^{\boldsymbol{\pi}}(S_2'; S_1, A_1 | S_2, A_2) \tag{47}$$

$$= \sum_{\boldsymbol{s}, \boldsymbol{a}, s_2' \in (S, A, S_2)} p^{\boldsymbol{\pi}}(\boldsymbol{s}, \boldsymbol{a}, s_2') \left[ Q_2^{\boldsymbol{\pi}}(\boldsymbol{s}, \boldsymbol{a}, s_2') - Q_{2|1}^{\boldsymbol{\pi}, *}(s_2, a_2, s_2') \right] \tag{48}$$

$$= \sum_{s_1, s_2, a_1, a_2} p^{\boldsymbol{\pi}}(s_1, s_2, a_1, a_2) (\tilde{r}_2(s_1, s_2, a_1, a_2) - \tilde{r}_2^{\boldsymbol{\pi}}(s_2, a_2) + \tag{49}$$

$$\gamma \sum_{s_2'} p(s_2' | s_1, s_2, a_1, a_2) (V_2^{\boldsymbol{\pi}}(s_2' | s_1, s_2, a_1, a_2) - V_2^{\boldsymbol{\pi}, *}(s_2' | s_2, a_2)) \tag{50}$$

$$= \sum_{s_1, s_2, a_1, a_2} p^{\boldsymbol{\pi}}(s_1, s_2, a_1, a_2) (\tilde{r}_2(s_1, s_2, a_1, a_2) - \tilde{r}_2^{\boldsymbol{\pi}}(s_2, a_2) + \tag{51}$$

$$\gamma \sum_{s_1', s_2'} T(s_1', s_2'|s_1, s_2, a_1, a_2)(1 - \frac{p^{\boldsymbol{\pi}}(s_2'|s_2, a_2)}{p(s_2'|s_1, s_2, a_1, a_2)})V_2^{\boldsymbol{\pi}}(s_1', s_2')) \text{ (Lemma 1)} \quad (52)$$

$$= \mathbb{E}_\tau \left[ \tilde{r}_2(\boldsymbol{s}, \boldsymbol{a}) - \tilde{r}_2^{\boldsymbol{\pi}}(s_2, a_2) + \gamma \left(1 - \frac{p^{\boldsymbol{\pi}}(s_2'|s_2, a_2)}{p(s_2'|\boldsymbol{s}, \boldsymbol{a})}\right) V_2^{\boldsymbol{\pi}}(\boldsymbol{s}') \right]. \quad (53)$$

$$\square$$

## B.3 CALCULATING GRADIENT OF VOI

In order to optimize $VoI$ with respect to the parameters of agent policy, in Sec. 3.2.1, we propose to use target function and get:

$$\nabla_{\theta_1} VoI_{2|1}^{\boldsymbol{\pi}}(S_2'; S_1, A_1|S_2, A_2) \approx \sum_{\boldsymbol{s}, \boldsymbol{a} \in (S, A)} (\nabla_{\theta_1} p^{\boldsymbol{\pi}}(\boldsymbol{s}, \boldsymbol{a})) [\tilde{r}_2(\boldsymbol{s}, \boldsymbol{a}) - \tilde{r}_2^-(s_2, a_2) +$$

$$\gamma \sum_{\boldsymbol{s}'} T(\boldsymbol{s}'|\boldsymbol{s}, \boldsymbol{a}) \left(1 - \frac{p^-(s_2'|s_2, a_2)}{p(s_2'|\boldsymbol{s}, \boldsymbol{a})}\right) V_2^-(s_1', s_2')].$$

$$(54)$$

We prove that $\sum_{\boldsymbol{s}, \boldsymbol{a}} (\nabla_{\theta_1} p^{\boldsymbol{\pi}}(\boldsymbol{s}, \boldsymbol{a})) \tilde{r}_2^-(s_2, a_2)$ is 0 in the following lemma.

**Lemma 2.**

$$\sum_{s_1, s_2, a_1, a_2} (\nabla_{\theta_1} p^{\boldsymbol{\pi}}(s_1, s_2, a_1, a_2)) \tilde{r}_2^-(s_2, a_2) = 0. \quad (55)$$

*Proof.* Similar to the way that policy gradient theorem was proved by Sutton et al. (2000),

$$\sum_{s_1, s_2, a_1, a_2} (\nabla_{\theta_1} p^{\boldsymbol{\pi}}(s_1, s_2, a_1, a_2)) \tilde{r}_2^-(s_2, a_2) \quad (56)$$

$$= \nabla_{\theta_1} \sum_{s_1, s_2, a_1, a_2} p^{\boldsymbol{\pi}}(s_1, s_2, a_1, a_2)\tilde{r}_2^-(s_2, a_2) \quad (57)$$

$$= \sum_{s_1^0, s_2^0} d_0(s_1^0, s_2^0) \prod_{t=0}^{\infty} (\nabla_{\theta_1} \boldsymbol{\pi}(a_1^t, a_2^t|s_1^t, s_2^t)) T(s_1^{t+1}, s_2^{t+1}|s_1^t, a_1^t, s_2^t, a_2^t)\tilde{r}_2^-(s_2^t, a_2^t) \quad (58)$$

$$= \sum_{s_1^0, s_2^0} d_0(s_1^0, s_2^0) \prod_{t=0}^{\infty} [\boldsymbol{\pi}(a_1^t, a_2^t|s_1^t, s_2^t) (\nabla_{\theta_1} \log \boldsymbol{\pi}(a_1^t, a_2^t|s_1^t, s_2^t)) \quad (59)$$

$$T(s_1^{t+1}, s_2^{t+1}|s_1^t, a_1^t, s_2^t, a_2^t)\tilde{r}_2^-(s_2^t, a_2^t)] \quad (60)$$

$$= \sum_{s_1, s_2, a_1, a_2} p^{\boldsymbol{\pi}}(s_1, s_2, a_1, a_2) (\nabla_{\theta_1} \log \boldsymbol{\pi}(a_1, a_2|s_1, s_2)) \tilde{r}_2^-(s_2, a_2) \quad (61)$$

$$= \sum_{s_1, s_2, a_1, a_2} p^{\boldsymbol{\pi}}(s_1, s_2, a_1, a_2) (\nabla_{\theta_1} \log \boldsymbol{\pi}(a_1|s_1, s_2)) \tilde{r}_2^-(s_2, a_2) \quad (62)$$

$$= \sum_{s_2, a_2} p^{\boldsymbol{\pi}}(s_2, a_2) \sum_{s_1, a_1} p^{\boldsymbol{\pi}}(s_1, a_1|s_2, a_2) (\nabla_{\theta_1} \log \boldsymbol{\pi}(a_1|s_1, s_2)) \tilde{r}_2^-(s_2, a_2) \quad (63)$$

$$= \sum_{s_2, a_2} p^{\boldsymbol{\pi}}(s_2, a_2)\tilde{r}_2^-(s_2, a_2) \sum_{s_1, a_1} p^{\boldsymbol{\pi}}(s_1, a_1|s_2, a_2) (\nabla_{\theta_1} \log \boldsymbol{\pi}(a_1|s_1, s_2)) \quad (64)$$

$$= \sum_{s_2, a_2} p^{\boldsymbol{\pi}}(s_2, a_2)\tilde{r}_2^-(s_2, a_2) \sum_{s_1, a_1} \frac{p^{\boldsymbol{\pi}}(s_1, a_1|s_2, a_2)}{\boldsymbol{\pi}(a_1|s_1, s_2)} (\nabla_{\theta_1} \boldsymbol{\pi}(a_1|s_1, s_2)) \quad (65)$$

$$= \sum_{s_2, a_2} p^{\boldsymbol{\pi}}(s_2, a_2)\tilde{r}_2^-(s_2, a_2) \sum_{s_1, a_1} \frac{p^{\boldsymbol{\pi}}(s_1|s_2, a_2)p^{\boldsymbol{\pi}}(a_1|s_1, s_2, a_2)}{\boldsymbol{\pi}(a_1|s_1, s_2)} (\nabla_{\theta_1} \boldsymbol{\pi}(a_1|s_1, s_2)) \quad (66)$$

$$= \sum_{s_2, a_2} p^{\boldsymbol{\pi}}(s_2, a_2)\tilde{r}_2^-(s_2, a_2) \sum_{s_1, a_1} \frac{p^{\boldsymbol{\pi}}(s_1|s_2, a_2)\boldsymbol{\pi}(a_1|s_1, s_2)}{\boldsymbol{\pi}(a_1|s_1, s_2)} (\nabla_{\theta_1} \boldsymbol{\pi}(a_1|s_1, s_2)) \quad (67)$$

$$= \sum_{s_2, a_2} p^{\boldsymbol{\pi}}(s_2, a_2) \tilde{r}_2^-(s_2, a_2) \sum_{s_1, a_1} p^{\boldsymbol{\pi}}(s_1|s_2, a_2) \left( \nabla_{\theta_1} \boldsymbol{\pi}(a_1|s_1, s_2) \right) \tag{68}$$

$$= \sum_{s_2, a_2} p^{\boldsymbol{\pi}}(s_2, a_2) \tilde{r}_2^-(s_2, a_2) \sum_{s_1} p^{\boldsymbol{\pi}}(s_1|s_2, a_2) \sum_{a_1} \left( \nabla_{\theta_1} \boldsymbol{\pi}(a_1|s_1, s_2) \right) \tag{69}$$

$$= \sum_{s_2, a_2} p^{\boldsymbol{\pi}}(s_2, a_2) \tilde{r}_2^-(s_2, a_2) \sum_{s_1} p^{\boldsymbol{\pi}}(s_1|s_2, a_2) \left( \nabla_{\theta_1} \sum_{a_1} \boldsymbol{\pi}(a_1|s_1, s_2) \right) \tag{70}$$

$$= \sum_{s_2, a_2} p^{\boldsymbol{\pi}}(s_2, a_2) \tilde{r}_2^-(s_2, a_2) \sum_{s_1} p^{\boldsymbol{\pi}}(s_1|s_2, a_2) \left( \nabla_{\theta_1} 1 \right) \tag{71}$$

$$= 0 \tag{72}$$

$\square$

### B.4 IMMEDIATE REWARD INFLUENCE

Similar to MI and $VoI$, we can define influence of agent 1 on the immediate rewards of agent 2 as:

$$RI_{2|1}^{\boldsymbol{\pi}}(S_2'; S_1, A_1 | S_2, A_2) = \sum_{\boldsymbol{s}, \boldsymbol{a} \in (S, A)} p^{\boldsymbol{\pi}}(\boldsymbol{s}, \boldsymbol{a}) [\tilde{r}_2(\boldsymbol{s}, \boldsymbol{a}) - \tilde{r}_2(s_2, a_2)]. \tag{73}$$

Use Lemma 2, we can get:

$$\nabla_{\theta_1} RI_{2|1}^{\boldsymbol{\pi}}(S_2'; S_1, A_1 | S_2, A_2) = \sum_{\boldsymbol{s}, \boldsymbol{a} \in (S, A)} \nabla_{\theta_1} (p^{\boldsymbol{\pi}}(\boldsymbol{s}, \boldsymbol{a})) \tilde{r}_2(\boldsymbol{s}, \boldsymbol{a}). \tag{74}$$

Now we have

$$\nabla_{\theta_1} J_{\theta_1}(t) \approx \left( \hat{R}_1^t - \hat{V}_1^{\boldsymbol{\pi}}(s_t) \right) \nabla_{\theta_1} \log \pi_{\theta_1}(a_1^t | s_1^t), \tag{75}$$

where $\hat{V}_1^{\boldsymbol{\pi}}(s_t)$ is an augmented value function of $\hat{R}_1^t = \sum_{t'=t}^{h} \hat{r}_1^{t'}$ and

$$\hat{r}_1^t = r^t + u_1^t + \beta u_2^t. \tag{76}$$

## C ESTIMATION OF CONDITIONAL PROBABILITIES

To quantify interdependence among exploration processes of agents, we use mutual information and value of interaction. Calculations of MI and VoI need estimation of $p(s_2'|s_2, a_2)$ and $p(s_2'|\boldsymbol{s}, \boldsymbol{a})$. In practice, we track the empirical frequencies $p_{emp}(s_2'|s_2, a_2)$ and $p_{emp}(s_2'|\boldsymbol{s}, \boldsymbol{a})$ and substitute them for the corresponding terms in Eq. 8 and 16.

Estimating $p_{emp}(s_2'|s_2, a_2)$ and $p_{emp}(s_2'|\boldsymbol{s}, \boldsymbol{a})$ is one obstacle to the scalability of our method, we now discuss how to solve this problem. When the state and action space is small, we can use hash table to implement Monte Carlo method (MC) for estimating the distributions accurately. In the MC sampling, we count from the samples the state frequencies $p(s_2'|s_2, a_2) \equiv \frac{N(s_2', s_2, a_2)}{N(s_2, a_2)}$ and $p(s_2'|\boldsymbol{s}, \boldsymbol{a}) \equiv \frac{N(s_2', s_1, s_2, a_1, a_2)}{N(s_1, s_2, a_1, a_2)}$, where $N(\cdot)$ is the number of times each state-action pair was visited during the learning process. When the problem space becomes large, MC consumes large memory in practice. As an alternative, we adopt variational inference (Fox & Roberts, 2012) to learn variational distributions $q_{\xi_1}(s_2'|s_2, a_2)$ and $q_{\xi_2}(s_2'|\boldsymbol{s}, \boldsymbol{a})$, parameterized via neural networks with parameters $\xi_1$ and $\xi_2$, by optimizing the evidence lower bound. In Fig. 6, we show the performance of EDTI estimated using variational inference and the changing of associated EDTI rewards on *Pass* during 9000 PPO updates. Variational inference introduces some noise in EDTI rewards estimation and thus requires slightly more steps to learn the true probability and the strategy. However, estimating using MC sampling consumes 1.6G memory to save the hash table with 100M items each agent while variational inference needs a three-layer fully connected network with 74800 parameters occupying about 0.60M memory. This results highlights the feasibility of estimating EITI and EDTI rewards using variational inference in problem with large state-action space.

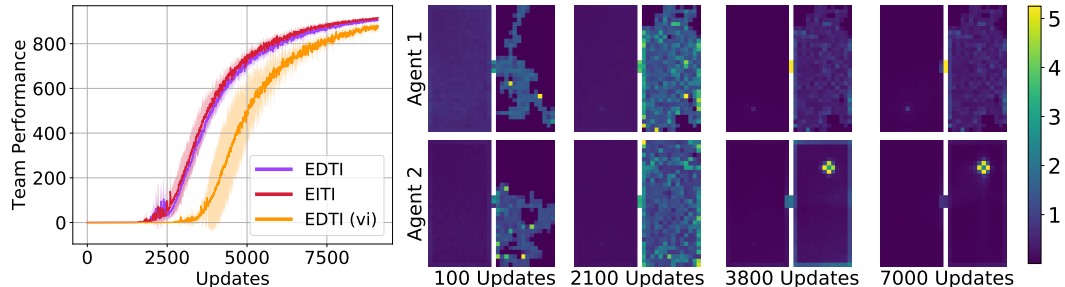

Figure 6: Left: Performance of EDTI (vi) (EDIT estimated using variational inference) compared with EITI and EDTI estimated using MC sampling. Others: Development of EDTI (vi) rewards during exploration process. Top row: EDTI (vi) rewards of agent 1; bottom row: EDTI (vi) rewards of agent 2.

Table 2: The scaling weights for different intrinsic reward terms in various tasks. $\beta_{\mathrm{T}}$ is the weight of term $T_1$ (see Table 1). $\beta_{\mathrm{int}}$ and $\beta_{\mathrm{ext}}$ are scaling factors to combine $r$ and $u_i$ in $\tilde{r}$. $u_{-i}$ in r_influence is scaled by $\beta_{\mathrm{r}}$ while $V_{-i}^{int}$ and $V_{-i}^{ext}$ in plusV are respectively scaled by $\beta_{\mathrm{int}}^{\mathrm{plusV}}$ and $\beta_{\mathrm{ext}}^{\mathrm{plusV}}$.

| Task | $\eta$ | $\beta_{\mathrm{T}}$ | $\beta_{\mathrm{int}}$ | $\beta_{\mathrm{ext}}$ | $\beta_{\mathrm{r}}$ | $\beta_{\mathrm{int}}^{\mathrm{plusV}}$ | $\beta_{\mathrm{ext}}^{\mathrm{plusV}}$ |
|---|---|---|---|---|---|---|---|
| *Pass* | 10. | 10 | 1. | 0.1 | 1. | 0.1 | 0.01 |
| *Secret-Room* | 10. | 10 | 1. | 0.1 | — | — | — |
| *Push-Box* | 1. | 100. | 100. | 0.1 | 0.1 | 0.1 | 0.01 |
| *Island* | 1. | 10 | 10. | 0.5 | 0.1 | 0.1 | 0.01 |
| *Large-Island* | 1. | 10 | 1. | 0.1 | 0.1 | 0.1 | 0.01 |

## D  IMPLEMENTATION DETAILS

### D.1  NETWORK ARCHITECTURE, HYPERPARAMETERS, AND INFRASTRUCTURE

We base our framework on OpenAI implementation of PPO2 (Dhariwal et al., 2017) and use its default parameters to carry out all the experiments. We train our models on an NVIDIA RTX 2080TI GPU using experience sampled from 32 parallel environments. We use visitation count to calculate the intrinsic reward, for its provable effectiveness (Azar et al., 2017; Jin et al., 2018). For all our methods and baselines, we use $\eta/\sqrt{N(s)}$ as the exploration bonus for $N(s)$-th visit to state $s$. Specific values of $\eta$ and scaling weights can be found in Table 2.

As for variational inference, the inference network is a 3-layer fully-connected network coupled with a 64-dimensional reparameterization estimator. ReLU is used as the activation function for the first two layers and the sum of negative log-likelihood and negative Evidence Lower Bound is used as loss. We use Adam optimizer (Kingma & Ba, 2014) with learning rate $1 \times 10^{-3}$ and batchsize 2048. To speed up the learning of variational distributions estimation, we equip the learning with proportional prioritized experience replay (Schaul et al., 2015).

### D.2  TASK STRUCTURE

In this section, we describe the detailed settings of the experimental tasks.

**Pass:** There are two agents and two switches to open the door in a $30 \times 30$ grid. Only when at least one of the switches are occupied will the door open. The agents need navigate from left to right and the team reward, which is 1000, is only provided when all agents reach the target zone. Agents can observe the position of another agents.

**Secret-Room:** This is an extension of the *Pass* task with 4 rooms and 4 switches locating in different rooms. The size of the grid is $25 \times 25$. When the left switch is occupied, all the three doors are open. And the three switches in each room on the right only control the door of its room. The agents need

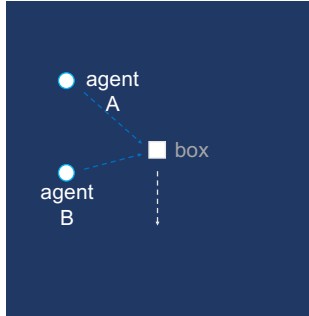 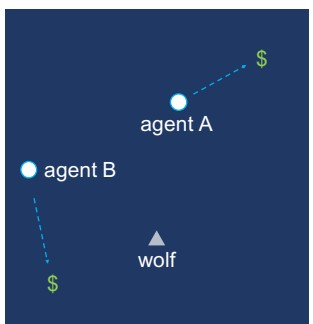 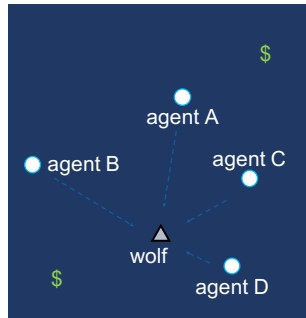

Figure 7: Task **Push-Box**, **Island**, and **Large-Island**

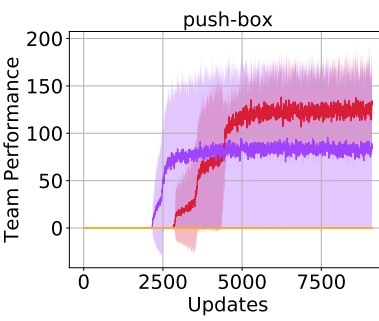 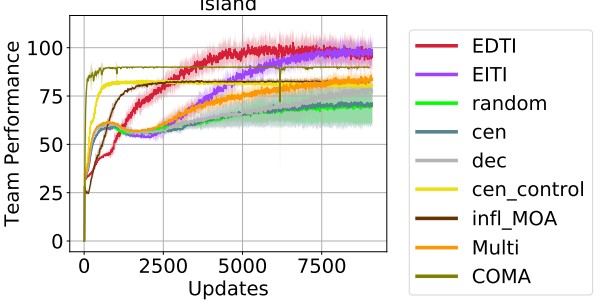

Figure 8: Comparison of our methods against baselines on *Push-Box* (left), *Island* (right).

to navigate towards the desired room (in light red of Fig. 1 middle) to achieve the extrinsic team reward 1000. Agents can observe the position of the other agents.

**Push-Box:** There are two agents and one box in a $15 \times 15$ grid. Agents need to push the box to the wall. However, the box is so heavy that only when two agents push it in the same direction at the same time can it be moved a grid. The only team reward, 1000, is given when the box is placed right against the wall. Agents can observe the coordinates of their teammate and the location of the box.

**Island:** A group of two agents are hunting for treasure on an island. However, a random walking beast may attack the agents when they are too close. The agents can also attack the beast within their attack range. This hurt doubles when more than one agent attack at the same time. Each agent has a maximum health of 5 and will lose $1/n$ health per step when there are $n$ agents within the attack range of the beast. *Island* is a modified version of the classic coordination scenario *Stag-Hunt* with local optimal, because finding each treasure (9 in total) will trigger a team reward of 10 but catching the beast gives a higher team reward of 300. Agents can observe the position and health of each other, and the coordinates of the beast. Fig. 9 shows the development of the probability of catching the beast and the averaged number of treasures found in an episode during 9000 PPO updates.

**Large-Island:** Settings are similar to that of *Island* but with more agents (4), more treasures (16), and a beast with more energy (16) and a higher reward (600) for being caught.

The horizon of one episode is set to 300 timesteps in all these tasks.

## E    COMPARISON WITH SINGLE-AGENT EXPLORATION METHODS

In this paper, we study the exploration problem in multi-agent settings from a decentralized perspective. Alternatively, exploration can be carried out in a centralized manner – treating agents as a joint one and using single-agent exploration algorithms. In this section, we compare our methods with centralized exploration strategies using RND (Burda et al., 2018) and EMI (Hyoungseok Kim, 2019), which are among the most cutting-edge exploration algorithms driven by curiosity and based on mutual information, respectively. We use codes published by their authors and carry out a modest

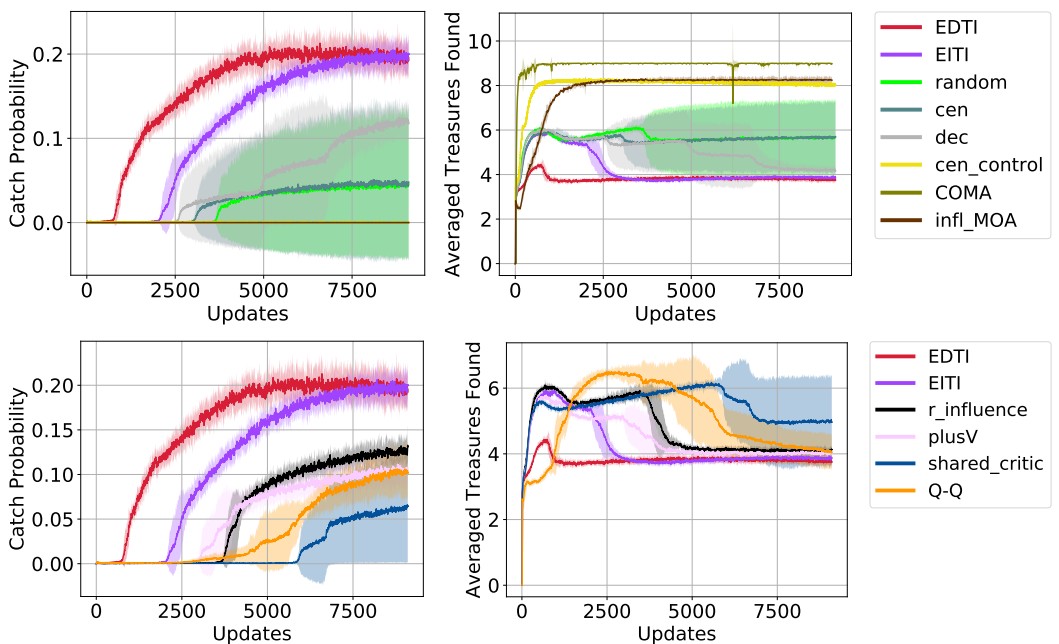

Figure 9: Comparison of our methods against baselines and ablations on *Island* in terms of the probability of catching the beast and the averaged treasures collected in an episode.

grid search over hyperparameters. For RND, we search intrinsic reward coefficient in the range of $[0.005, 1.0]$ and extrinsic reward coefficient in range $[0.05, 2.0]$. For EMI, we test difference combinations of loss weights. Results averaged over four random seeds with the best found parameters are shown below.

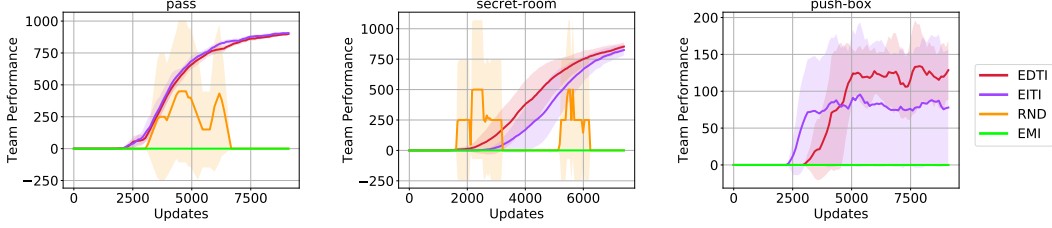

Figure 10: Comparison of our methods against centralized single-agent exploration algorithms on *Pass* (left), *Secret-Room* (middle), and *Push-Box* (right).

Performance comparisons on problems of *Pass*, *Secret-Room*, and *Push-Box* are illustrated in Fig. 10. We can observe that our methods significantly outperform centralized exploration strategies using RND or EMI. To better understand this observation, we plot visitation heatmaps over time for RND and EMI, respectively, in Fig. 11 and 12.

Fig. 11 shows visitation heatmaps of RND on the *Pass* problem. From Fig. 11 (b), we can see that RND seems finding good policies for agents to pass the door in the first 4671 updates. However, agents' policies seem to collapse quickly after that and their visits scatter around rooms again, which explains its learning curve in Fig. 10. From the evolution of its visitation heatmaps, we hypothesize that after visiting the center of the room for many times, agents' curiosity models overfit on a particular set of states and they start to be curious about the relatively unfamiliar transition dynamics around the wall. As the result, the RND intrinsic reward drags the agents to the walls, as shown in Fig. 11(c) and (d), and their performance quickly drops within several updates (i.e., update 4671-4677 shown by Fig. 11(b-d)). After a while, agents then leave from the walls and visit around in the

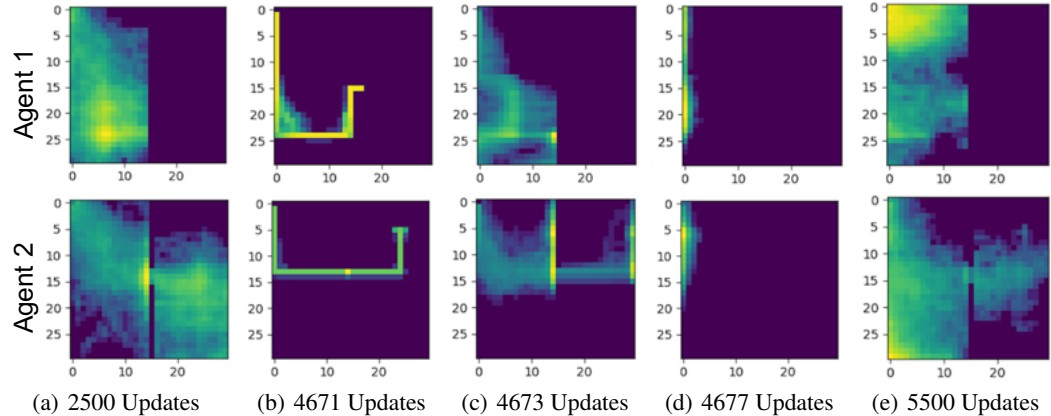

Figure 11: Visitation heatmap of RND agents on *Pass* of most recent $1k$ episodes. The brighter the yellow color, the higher the visitation frequency. Top: agent 1, bottom: agent 2.

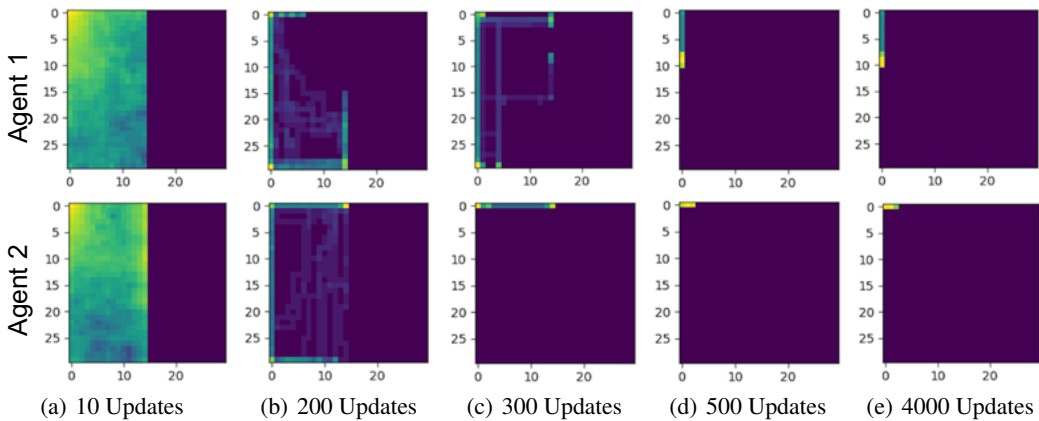

Figure 12: Visitation heatmap in most recent $1k$ episodes of EMI agents on *Pass*. The brighter the yellow color, the higher the visitation frequency. Top: agent 1, bottom: agent 2.

room again, as shown in Fig. 11(e). The whole exploration process repeated. Similar behaviors are also observed on the *Secret-Room* problem.

We also analyze the exploration behaviors of EMI agents on *Pass*, as illustrated by visitation heatmaps in Fig. 12. EMI tends to explore the state-action pairs where the transition dynamics is relatively complex, such as the edges and corners of the room (Fig. 12(a-c)). For problems where these state-action pairs do not lead to goals, EMI is not very effective. As the (centralized) transition dynamics of the *Pass* problem is relatively simple, EMI intrinsic reward quickly diminishes, which results in the behaviors of agents keeping unchanged after 500 updates (Fig. 12(d-e)).

In summary, centralized single-agent exploration methods encode some heuristics to facilitate exploration, but they typically do not place a great emphasis on interactions among agents and are thus not very efficient for multi-agent exploration with sparse interactions.

