# OpenReview forum: "Influence-Based Multi-Agent Exploration"
_ICLR.cc/2020/Conference — Accept (Spotlight)_

### Official Review · AnonReviewer3 · 2019-10-20
**Official Blind Review #3**

**Rating:** 8

**Review:**

This paper studies the problem of designing effective exploration strategies in multi-agent domains. The key idea is to define one agent's exploration in terms of its interactions with other agents. This leads to two auxiliary exploration objectives, which measure how one agent's actions affect the dynamics and value of another agent's actions. The paper does an admirable job comparing the proposed method against a number of baselines, where the proposed method performs significantly better. Visualizations and ablation experiments nicely illustrate the contributions of various components of the method.

I am leaning towards accepting the paper. To the best of my knowledge, the broad idea of applying information theory to multi-agent exploration, in addition to the specific instantiation described in the paper, is novel. I expect that this paper will encourage future work to explore more problems in this area. The experiments are quite thorough. My main reservation is a lack of comparisons to single agent exploration methods. As noted in Section 3, we can view multi-agent domains as just a special type of single agent domain. How would curiosity-based exploration, such as [Burda 2018, Pathak 17], or mutual information-based exploration, such as [Gregor 16, Eysenbach 18, Achaim 18], compare to the proposed method?

I have a few reservations about the clarity of presentation, but I think those are easily addressed. My remaining concern is that the results are on somewhat toy tasks, but I think that is par for this area of research.

Overall, I would strongly argue for accepting this paper if comparisons to single-agent exploration methods were added. I would consider decreasing my review if another reviewer found quite similar prior work, or if significant bugs were found in the mathematical derivation (I have not carefully checked all the proofs in the appendix.).

Minor comments
* "transition-dependent" -- what does this mean?
* "while tend" -- missing a subject
* "struggle in many real-world scenarios with sparse rewards" -- please add a citation
* "intrinsic value function of agent i, I_{-i|i}^\pi is \beta > 0 is a weighting" -- I think part of this sentence was accidentally deleted.
* Eq 5: What is the difference between I and MI?
* "We call …" -- What is the a_2^V0I^\pi_{-i|i} term?
* Nitpick: Use `` for the start of quotes
* Appendix B1: How is Eq 22 obtained from Eq 21?

-------------------UPDATE AFTER AUTHOR RESPONSE---------------------
The authors have done a great job address my two concerns (similarity to prior work and empirical comparisons with single-agent exploration). I therefore increase my vote to "accept."

**Experience Assessment:**

I have read many papers in this area.

**Review Assessment: Checking Correctness Of Derivations And Theory:**

I assessed the sensibility of the derivations and theory.

**Review Assessment: Checking Correctness Of Experiments:**

I assessed the sensibility of the experiments.

**Review Assessment: Thoroughness In Paper Reading:**

I read the paper at least twice and used my best judgement in assessing the paper.

---

> ### Author Response · Authors · 2019-11-14
> **Thanks for your constructive comments. We have added experiments for comparing our methods to single-agent exploration methods and corresponding analyses in the updated version.**
>
> Thanks for your constructive comments. We have added experiments for comparing our methods to single-agent exploration methods and corresponding analyses in the updated version. Here we provide explanations to your questions.
>
> Q1: Comparison to single-agent exploration methods.
> A1: As requested, we compare our methods with RND [Burda et al., ICLR 2019a] and EMI (exploration with mutual information) [Kim et al., ICML 2019]. We choose these two methods because they are among the most cutting-edge curiosity-driven and mutual information-based exploration algorithms, respectively.
>
> So far, we have carried out a modest grid search over some hyperparameters for both RND (coefficients of intrinsic and extrinsic reward) and EMI (loss weights) and provided results with their fine-tuned parameters in a new Appendix section (Appendix E, page 21-23). Briefly speaking, our approaches significantly outperform RND and EMI in problem settings illustrated in this paper. To better understand this observation, we plotted and analyzed agents’ visitation heatmaps over time and found they tended to focus exploration on state-action pairs where the environment transition dynamics is complex, instead of exploring state-action pairs which can lead to interactions potentially with external rewards. Please refer to detailed results and discussions in the updated version.
>
> Q2: About minor comments.
> Q: What does “transition-dependent” mean?
> A: Transition-dependent means that the transition dynamics of one agent is dependent on states and actions of other agents, e.g., if $p(s_1’, s_2’ | s_1, a_1, s_2, a_2) \ne p(s_1’ | s_1, a_1) p(s_2’ | s_2, a_2)$, then agent 1 and 2 are transition dependent.
>
> Q: "struggle in many real-world scenarios with sparse rewards" -- please add a citation
> A: We added a citation to [Burda et al., ICLR 2019b], which makes a similar comment and carries out comprehensive experiments on sparse reward tasks.
>
> Q: "intrinsic value function of agent $i$, $I_{-i|i}^\pi$ is $\beta > 0$ is a weighting" -- I think part of this sentence was accidentally deleted.
> A: Thank you for pointing out this typo. In the updated version, we have corrected it, which should be “$I_{-i|i}^\pi$ is the influence value, $\beta > 0$ is a weighting term”.
>
> Q: Eq 5: What is the difference between $I$ and MI?
> A: The notation $I$ represents the influence value and we propose two methods to instantiate it, i.e., MI and VoI, respectively.
>
> Q: "We call …" -- What is the a_2^VoI^\pi_{-i|i} term?
> A: Thank you for pointing out this typo and the VoI term should not be included. We have corrected it in the new version. The denominator should be $p(s_2^{t+1} | s_1^t, s_2^t, a_1^t, a_2^t)$.
>
> Q: Appendix B1: How is Eq 22 obtained from Eq 21?
> A: The partial derivative of the numerator in Eq. 21 is 0, because $p(s_2’ | s_1, s_2, a_1, a_2)$ is decided by the transition function of the factored multi-agent MDP and is independent of $\theta_1$ (the policy parameters of agent 1). Therefore, Eq. 22 only contains the partial derivative of the denominator of the log term.
>
>
> [Burda et al., ICLR 2019a] Burda, Y., Edwards, H., Storkey, A. and Klimov, O., 2018. Exploration by random network distillation. In Proceedings of the Seventh International Conference on Learning Representations.
>
> [Kim et al., ICML 2019] Kim, H., Kim, J., Jeong, Y., Levine, S. and Song, H.O., 2018. EMI: Exploration with mutual information. In Proceedings of the 36th International Conference on Machine Learning (Vol. 97, pp. 3360-3369).
>
> [Burda et al., ICLR 2019b] Burda, Y., Edwards, H., Pathak, D., Storkey, A., Darrell, T., & Efros, A. A. (2018). Large-scale study of curiosity-driven learning. In Proceedings of the Seventh International Conference on Learning Representations.

---

### Official Review · AnonReviewer1 · 2019-10-21
**Official Blind Review #1**

**Rating:** 6

**Review:**

Update: I thank the authors for their response and I will maintain my score, my main hesitation being the overall clarity and readability of the paper.

Summary:
This paper proposes the use of two intrinsic rewards for exploration in MARL settings. The first one is an information-theoretic influence (EITI) bonus and a decision-theoretic influence (EDTI)  bonus. EITI uses mutual information to capture the influence of one agent on the transition dynamics of others,  while EDTI uses an intrinsic reward called Value of Interaction (VoI) to quantify the influence of one agent’s behavior on expected returns of other agents.

Main Comments:
Overall, I think this paper would be a good contribution for ICLR 2020 and I lean towards accepting it. The experimental section is thorough, the authors include relevant ablations, baselines and popular algorithms used in MARL settings. The use of the decision-theoretic influence is novel as far as I can tell and it also seems to be quite effective on the tasks used for evaluation. Although the method uses a series of approximations and assumptions, I believe most of them are clearly stated and fairly well-motivated (plus they are not very far from those of other recent work in the deep MARL literature). I also appreciated the fact that the authors explicitly derived the main mathematical results used in the paper.

I only have some minor comments and questions regarding some assumptions and notation.

I  also encourage the authors to proof-read the paper as some parts of it are a bit hard to follow. I would very much like to see a more an edited version of this paper with more precise language.

Can you discuss in more detail the difference between EITI and the intrinsic reward based on social influence used in Jacques et al. (2018)? They seem to be quite similar conceptually and the related work part related to this is rather vague. Please clarify the distinction.

Minor Questions / Comments:
There are a few typos throughout the paper:

1. On page 3 after equation (3), I believe part of the sentence that should describe the I term in the equation is missing.

2. The phrase right after equation (16) which defines the EDTI reward does not seems to not match the  above expression. Can you please explain why the transition would be conditioned on the influence term? While reviewing, I’ve been assuming this was just a mistake in writing but please double check and clarify.

3. On page 4, after equation (8), you refer to  a_1, a_1 and s_2’ which do not appear in the above equation. Can you please use the same notation or motivate your choice for referring to those variables instead?




**Experience Assessment:**

I have published one or two papers in this area.

**Review Assessment: Checking Correctness Of Derivations And Theory:**

I assessed the sensibility of the derivations and theory.

**Review Assessment: Checking Correctness Of Experiments:**

I assessed the sensibility of the experiments.

**Review Assessment: Thoroughness In Paper Reading:**

I read the paper at least twice and used my best judgement in assessing the paper.

---

> ### Author Response · Authors · 2019-11-14
> **Thanks for your careful reading and thoughtful comments. Here we provide some feedback to your comments.**
>
> Thanks for your careful reading and thoughtful comments. Here we provide some feedback to your comments.
>
> Q1: A more edited version of this paper with more precise language is expected.
> A1: We have proofread the manuscript and made modifications to improve our presentation. We would appreciate it if you have more comments about the readability.
>
> Q2: Difference between EITI and the intrinsic reward based on social influence [Jaques et al., ICML 2019].
> A2: The differences between EITI and social influence lie in both definition and optimization. (1) Definition. Social influence is defined to be the influence of one agent on the policies of other agents, while EITI measures the influence of one agent on the transition dynamics of other agents. Accompanying this distinction, EITI considers both states and actions to measure the influence while social influence quantifies the influence of actions without considering state information, which is critical for exploration; (2) Optimization. In EITI, to explicitly maximize mutual information, we add it as a regularizer to the learning objective and derive the gradients in the policy gradient framework. In contrast, social influence reward is added to the immediate environmental reward, and is used to train the RL algorithms.
>
> We have also updated the relating discussions in Sec. 5 of the new version of our manuscript.
>
> Q3: About the typos.
> A3: Thank you for pointing out these typos. We have corrected them in the updated version.
>
> [Jaques et al., ICML 2019] Jaques, N., Lazaridou, A., Hughes, E., Gulcehre, C., Ortega, P., Strouse, D., Leibo, J.Z. and De Freitas, N., 2019, May. Social Influence as Intrinsic Motivation for Multi-Agent Deep Reinforcement Learning. In International Conference on Machine Learning (pp. 3040-3049).

---

> > ### Public Comment · ~Wenshuai_Zhao1 · 2021-08-31
> > **Question on A2: how does the influence reward work as a regularizer rather than just auxiliary reward?**
> >
> > Hi, good work! I think the way you restate in Answer 2 (A2) is very interesting that you treat the influence reward as a regularizer to optimize but not just added immediate reward as the 'social influence' does.
> >
> > Equation (1) illustrates this very well. However, in Equation (16), the three rewards are still merged together as the social influence does. So, can you elaborate on this in more detail?
> >
> > Maybe there are still other readers having the same confusion as me. So, your explanation will be very helpful.
> >
> > Thanks!

---

### Official Review · AnonReviewer2 · 2019-10-25
**Official Blind Review #2**

**Rating:** 6

**Review:**

This paper proposes methods for incentivizing exploration in multi-agent RL.  There are two approaches that are proposed, both framed as influence maximization (of either the state transitions or the decisions of the other agents).  The scaling to multiple agents is done via decomposing to pairwise interactions. This influence objective is the appended to the standard intrinsic motivation objective for single agent RL.

The proposed approaches are pretty elegant, and in a sense seem fundamental.  I'm not an expert in this particular area, so I don't know how novel these ideas are.  (See related work comments below.)

The empirical results seem quite strong, although (being a a non-expert), I can't tell if they're constructed to be good for the proposed approaches.  There isn't much discussion of limitations and/or experiments breaking the proposed approach.

I found the related work discussion a bit incomplete.  Can the authors comment directly on related MARL work, such as Foerster et al., AAAI 2018?  What are the specific points of contrast?

**Experience Assessment:**

I have read many papers in this area.

**Review Assessment: Checking Correctness Of Derivations And Theory:**

I carefully checked the derivations and theory.

**Review Assessment: Checking Correctness Of Experiments:**

I assessed the sensibility of the experiments.

**Review Assessment: Thoroughness In Paper Reading:**

I read the paper at least twice and used my best judgement in assessing the paper.

---

> ### Author Response · Authors · 2019-11-14
> **Thanks for your detailed comments. Here we provide clarification for your comments.**
>
> Thanks for your detailed comments. Here we provide clarification for your comments.
>
> Q1: How novel our ideas are?
> A1: To our best knowledge, this paper is the first work that proposes the general idea of introducing influence between agents into multi-agent exploration (as discussed in the related work part). In addition, we present two original instantiations for quantifying influence: information-theoretic measure based on mutual information and decision-theoretic measure based on counterfactual value, and also show how to optimize them in the policy gradient framework.
>
> Q2: Discussions on limitations.
> A2: We discuss the main limitation of our methods — the scalability issue — in Appendix C. We also propose a variational inference approach there to alleviating this problem and do some experiments to prove the feasibility of this method.
>
> As discussed in Section 3.3, for scenarios with tight-coupled interaction, influences involving more than two agents are approximately decomposed into pairwise interactions, which may be inaccurate and is another potential limitation of our method.
>
> Q3: Comments on related MARL works such as COMA.
> A3: COMA [Foerster et al., AAAI 2018] shares some similarities with EDTI in that both of them use the idea of counterfactual formulations. However, they are quite different in terms of definition and optimization: (1) conceptually, EDTI is proposed to measure the influence of one agent on the value functions of other agents, while COMA aims at quantifying individual contribution to the team value; (2) EDTI is defined on counterfactual Q values over state-action pairs of other agents given its own state-action pair while COMA uses the counterfactual Q value just over its own action without considering state information, which is critical for exploration; (3) we explicitly derive the gradients for optimizing EDTI influence for coordinated exploration in the policy gradient framework, which provides more accurate feedback, while COMA uses the counterfactual Q value as a critic.
>
> Apart from COMA, our work is also related to influence-based abstraction [Oliehoek et al., 2012, de Castro et al., 2019] and social influence [Jaques et al., ICML 2019]. We have added discussions in Sec. 5 of the updated version of our manuscript.
>
> [Foerster et al., AAAI 2018] Foerster, J.N., Farquhar, G., Afouras, T., Nardelli, N. and Whiteson, S., 2018, April. Counterfactual multi-agent policy gradients. In Thirty-Second AAAI Conference on Artificial Intelligence.
>
> [de Castro et al., 2019] de Castro, M.S., Congeduti, E., Starre, R.A., Czechowski, A. and Oliehoek, F.A., 2019, May. Influence-based abstraction in deep reinforcement learning. In Adaptive, learning agents workshop (Vol. 34).
>
> [Oliehoek et al., 2012] Oliehoek, F.A., Witwicki, S.J. and Kaelbling, L.P., 2012, July. Influence-based abstraction for multiagent systems. In Twenty-Sixth AAAI Conference on Artificial Intelligence.
>
> [Jaques et al., ICML 2019] Jaques, N., Lazaridou, A., Hughes, E., Gulcehre, C., Ortega, P., Strouse, D., Leibo, J.Z. and De Freitas, N., 2019, May. Social Influence as Intrinsic Motivation for Multi-Agent Deep Reinforcement Learning. In International Conference on Machine Learning (pp. 3040-3049).

---

### Public Comment · ~Frans_Oliehoek1 · 2019-10-03
**about the link to "Influence-based Abstraction" and a resulting question**

Dear authors,

Thank you for an interesting submission. I would like to bring to your attention a body of related work [1,2,3] (my refs below), that I think you missed.

(Sorry, I hate to be that 'you forgot about my paper' person, and certainly for a paper that clearly tries to cite all related work. Still, in this case, I think my posting here is justified. But, let me stress that I think your paper certainly has a different, novel, angle on exploration that I find very interesting. However, the link to our work does raise one important question about your approach.)

Together with coauthors, and building upon a string of previous papers (e.g. [2], see [1] for further pointers), I have developed a theory of what we call "influence-based abstraction" (IBA) [1] for abstraction in multiagent settings (factored Dec-POMDPs). Essentially our work analyses what an agent needs to remember about the past, to correctly predict the "influence sources" that will affect its local decision making. It gives a definition of influence, dubbed an "influence point" that perfectly captures all information the agent needs to take local actions without any loss in value. Recently, we also extended some of these insights to the deep RL domain [3].

The setting described in section 3 in this paper (dubbed "(agent-wise) factored Dec-MDP", not multiagent MDP, in the literature [2], see my book [4] for an overview of these models), falls squarely within the description of IBA. As such, I think it would be enlightening to make this connection.

In particular, from the definition of influence in our work (definition 4 and 5 in the AAAI paper) it directly follows that, without any assumptions on the structure, in order to predict the other agents an agent needs to condition on the full history of observations. (As a side note: this also implies that agents really should use history dependent policies.)

This leads to what I think is an important question:

how is the term
p^\pi( s_2' | s_2, a_2 )
actually defined?
(and similar for p^\pi( s_1*, a_1* | s_2, a_2 ) in the second approach).

From the theory of IBA, we know that these type of distributions depend on the history, and therefore the time step. It is not clear that averaging over time-steps is a sensible solution (which is de facto what is being done when keeping empirical counts, as explained in appendix C): it could be that the probabilities flip-flop between different stages, and averaging would give a completely wrong prediction?

This also seem to be an issue in (6) and the proof in appendix B: suppose that p^\pi( s_2' | s_2, a_2 ) indeed is time dependent, then these definitions and substitutions in the derivation are insufficient, it seems?

(I suppose, another option could be try and build these notions around the concept of state state distributions, but that would need additional assumptions, and still would pose its problems in the learning setup where such a stationary distribution is never reached before the learning has converged?)

Looking forward to hearing your thoughts on this.

With kind regards,
-Frans Oliehoek



[1] F. A. Oliehoek, S. Witwicki, and L. P. Kaelbling. Influence-based abstraction for multiagent systems. In Proceedings of the Twenty-Sixth AAAI Conference on Artificial Intelligence, pages 1422–1428, July 2012
* Also see a recent longer version: https://arxiv.org/abs/1907.09278

[2] R. Becker, S. Zilberstein, V. Lesser, and C. V. Goldman. Transition-independent decentralized Markov decision processes. In Proceedings of the International Conference on Autonomous Agents and Multiagent Systems, pages 41–48, 2003.

[3] Miguel Suau de Castro, Elena Congeduti, Rolf A.N. Starre, Aleksander Czechowski, and Frans A. Oliehoek. Influence-Based Abstraction in Deep Reinforcement Learning. In Proceedings of the AAMAS Workshop on Adaptive Learning Agents (ALA), May 2019.

[4] Frans A. Oliehoek and Christopher Amato. A Concise Introduction to Decentralized POMDPs, SpringerBriefs in Intelligent Systems, Springer, May 2016.

---

> ### Author Response · Authors · 2019-10-06
> **Clarification of the questions.**
>
> Dear Frans,
>
> Thank you for your interest in our work and thoughtful comments. We appreciate that you point out some interesting related works, for which we will include discussions in the next version. Here we would like to discuss the differences and clarify your questions.
>
> First, as you have mentioned in your comments, the problem settings are different. We study the exploration problem of multi-agent reinforcement learning in factored Dec-MDP. We find that encouraging agents to exert influence on the exploration processes of other agents facilitates coordinated exploration. Therefore, we try to maximize the influence among agents. To this end, we quantify the influence using mutual information (measuring influence between transition probabilities) and value of interaction (measuring influence between value functions, including intrinsic rewards). Your work on IBA identifies influence points in the observation history that are necessary for the decision-making of one agent.
>
> The second difference lies in techniques — our definitions are based on stationary state distributions for a given joint policy. We will now explain why we choose stationary distributions and how to make the estimations stable.
>
> To measure the influence of an agent on the exploration processes of other agents, we define mutual information (MI) and value of interaction (VoI). Our discussions and definitions in Section 4 focus on infinite-horizon cases. In multi-agent settings, individual policies are generally based on local action-observation history. However, in infinite-horizon cases, this will render measuring the influence between agents challenging. To make measuring influence between agents tractable, we propose a new approach in this paper. We exploit the stationary distribution of the global state and the joint action, which we assume to exist for a given joint policy (many works studying multi-agent policy gradients make similar arguments, such as [1, 2]), to define MI and VoI. With this stationary distribution for a given joint policy \pi, distributions like p^\pi(s_2' | s_2, a_2) and p^\pi( s_1*, a_1* | s_2, a_2 ) can be obtained by marginalizing over some variables.
>
> As for your concern about the stability of the estimations of these probabilities, in our experiments, we 1) collect a large number of steps (2k steps in each of 32 parallel environments) between policy updates; 2) slowly update target functions (e.g., p^-(s_2’ | s_2, a_2) in Eq. 13) which are used to calculate the corresponding gradients. Some papers have used similar techniques to approximately estimate conditional probabilities related to stationary distributions [3].
>
> We also would like to mention that we adopt the paradigm of centralized training with decentralized execution, and global state information and global value functions are only used during the training. In the decentralized execution, individual policies are based on local action-observation history.
>
> Thanks again for your comments,
> -Authors
>
>
> [1] Foerster, J.N., Farquhar, G., Afouras, T., Nardelli, N. and Whiteson, S., 2018, April. Counterfactual multi-agent policy gradients. In Thirty-Second AAAI Conference on Artificial Intelligence.
>
> [2] Lowe, R., Wu, Y., Tamar, A., Harb, J., Abbeel, O.P. and Mordatch, I., 2017. Multi-agent actor-critic for mixed cooperative-competitive environments. In Advances in Neural Information Processing Systems (pp. 6379-6390).
>
> [3] Strouse, D.J., Kleiman-Weiner, M., Tenenbaum, J., Botvinick, M. and Schwab, D.J., 2018. Learning to share and hide intentions using information regularization. In Advances in Neural Information Processing Systems (pp. 10249-10259).

---

> > ### Public Comment · ~Frans_Oliehoek1 · 2019-10-09
> > **thanks for clarification**
> >
> > Dear Authors,
> >
> > Thank you for your response, this is very helpful.
> >
> > I certainly agree to the differences in goal that you point out, and indeed the idea that somehow maximizing the influence could lead to better exploration because this "encourage[s] interactions" is interesting, and as far as I can tell novel.
> >
> > Also spelling out the assumptions is very helpful. If I understand correctly:
> > 1) you assume that just using the last local state is hopefully good enough.
> > 2) you assume that the stationary distribution of the global state and the joint action exists
> >
> > I think the former may make sense (but could make sense to try and validate).
> >
> > The latter is in my view more problematic: even though I agree that, for a *given joint policy* (that is a function of last observation only), this may be a justifyiable assumption (assuming ergodicity).
> >
> > However, during learning no such  joint policy is given: in contrast, the agents are constantly changing their policies. Only when the learning has ended (if/when the policies of the agents have converged), will the assumption finally be satisfied. However, the proposed methods employ the assumption to enable faster convergence, i.e., during the non-convergent phase. As such, the theory seems to be build on an assumption that pertinently does not hold? (And, certainly, it might be the case that other papers make similar mistakes.)
> >
> > Bottom line here is that I think the proper way to capture the influence exerted by a non-stationary learner, is via a concept of influence that aims to capture those non-stationarities, like (there may be others) the concept of influence point that we proposed.
> >
> > This of course does not preclude more approximate methods from possibly being effective too. This certainly seems to be supported by the experiments presented here. Perhaps the big open research question is why this seems to work so well.
> >
> > I would be very interested in discussing this further, but should stop here since this is drifting away from the current paper. If you also would be interested in discussing further feel free to contact me.
> >
> > With kind regards,
> > -Frans

---

> > > ### Author Response · Authors · 2019-10-15
> > > **We are glad that our previous reply has clarified some of your questions. Explanations about the estimation problem.**
> > >
> > > Dear Frans,
> > >
> > > Thanks for your thoughtful comments again. We are glad that our previous reply has clarified some of your questions. Here we would like to provide explanations about the estimation of probabilities like $p^\pi(s_2’ | s_2, a_2)$.
> > >
> > > We agree with you that \pi is changing during learning and we are approximately estimating the probabilities. As you have pointed out, such a proper approximation can be effective. In fact, we note that such approximation is quite common (and empirically effective) in reinforcement learning. For example, policy gradient theorem uses Q^\pi in its updates [1]. Similar to our p^\pi(s_2’ | s_2, a_2), Q^\pi is also estimated while \pi is changing from time to time.
> > >
> > > We also would like to highlight some techniques borrowed from deep reinforcement learning toolkit that we find useful in robustly estimating probabilities like p^\pi(s_2’ | s_2, a_2): (1) more samples between policy updates collected from parallel running environments; (2) using a moving average to update a target estimation [2, 3, 4].
> > >
> > > We are trying to capture the influence among agents given a joint policy. Another perspective is to analyze how agents influence the evolving of policies of other agents. We think the latter one is precise, as you have done in your work, but such precision is somewhat costly in practice.
> > >
> > > Please feel free to contact us if you have any further questions.
> > >
> > > Best,
> > >
> > > Authors
> > >
> > >
> > >
> > > [1] Sutton, Richard S., David A. McAllester, Satinder P. Singh, and Yishay Mansour. "Policy gradient methods for reinforcement learning with function approximation." In Advances in neural information processing systems, pp. 1057-1063. 2000.
> > >
> > > [2] Mnih, V., Kavukcuoglu, K., Silver, D., Rusu, A.A., Veness, J., Bellemare, M.G., Graves, A., Riedmiller, M., Fidjeland, A.K., Ostrovski, G. and Petersen, S., 2015. Human-level control through deep reinforcement learning. Nature, 518(7540), p.529.
> > >
> > > [3] Lillicrap, T.P., Hunt, J.J., Pritzel, A., Heess, N., Erez, T., Tassa, Y., Silver, D. and Wierstra, D., 2015. Continuous control with deep reinforcement learning. International Conference on Representation Learning Representations, 2016.
> > >
> > > [4] Haarnoja, T., Zhou, A., Abbeel, P. and Levine, S., 2018, July. Soft Actor-Critic: Off-Policy Maximum Entropy Deep Reinforcement Learning with a Stochastic Actor. In International Conference on Machine Learning (pp. 1856-1865).

---

### Author Response · Authors · 2019-10-15
**Sorry for two typos.**

(1) In the second line below Eq. 1, page 3:

“$I^\mathbf{\pi}_{-i | i}$ is $\beta > 0$ is a weighting term.”
 ==>
“$I^\mathbf{\pi}_{-i | i}$ is the influence value, and $\beta > 0$ is a weighting term.”

(2) In the definition of EDTI reward in the first line below Eq. 16, page 6, the denominator should be  $p(s_2^{t+1} | s_1^t, s_2^t, a_1^t, a_2^t)$.

---

### Author Response · Authors · 2019-11-14
**General Response to Reviewers' Comments**

We thank all the reviewers for their efforts and helpful comments. As suggested, we have refined the paper, including additional experiments of comparison to single-agent exploration methods, adding discussions to additional related works, and improving its presentation.

---

### Decision · Program_Chairs · 2019-12-19

**Decision:**

Accept (Spotlight)

**Comment:**

The paper presents a new take on exploration in multi-agent reinforcement learning settings, and presents two approaches, one motivated by information theoretic, the other by decision theoretic influence on other agents. Reviewers consider the proposed approach "pretty elegant, and in a sense seem fundamental", the experimental section "thorough", and expect the work to "encourage future work to explore more problems in this area". Several questions were raised, especially regarding related work, comparison to single agent exploration approaches, and several clarifying questions. These were largely addressed by the authors, resulting in a strong submission with valuable contributions.